# LANGUAGE-IMAGE MODELS WITH 3D UNDERSTANDING

**Jang Hyun Cho**[1,*]   **Boris Ivanovic**[2]   **Yulong Cao**[2]   **Edward Schmerling**[2]   **Yue Wang**[2]

**Xinshuo Weng**[2]   **Boyi Li**[2]   **Yurong You**[2]   **Philipp Krähenbühl**[1,†]   **Yan Wang**[2,†]

**Marco Pavone**[2,†]

[1] University of Texas at Austin    [2] NVIDIA

## ABSTRACT

Multi-modal large language models (MLLMs) have shown incredible capabilities in a variety of 2D vision and language tasks. We extend MLLMs' perceptual capabilities to ground and reason about images in 3-dimensional space. To that end, we first develop a large-scale pretraining dataset for 2D and 3D called LV3D by combining multiple existing 2D and 3D recognition datasets under a common task formulation: as multi-turn question-answering. Next, we introduce a new MLLM named **CUBE-LLM** and pre-train it on LV3D. We show that pure data scaling makes a strong 3D perception capability without 3D-specific architectural design or training objectives. **CUBE-LLM** exhibits intriguing properties similar to LLMs: (1) **CUBE-LLM** can apply chain-of-thought prompting to improve 3D understanding from 2D context information. (2) **CUBE-LLM** can follow complex and diverse instructions and adapt to versatile input and output formats. (3) **CUBE-LLM** can be visually prompted such as 2D box or a set of candidate 3D boxes from specialists. Our experiments on outdoor benchmarks demonstrate that **CUBE-LLM** significantly outperforms existing baselines by 21.3 points of $AP_{BEV}$ on the Talk2Car dataset for 3D grounded reasoning and 17.7 points on the DriveLM dataset for complex reasoning about driving scenarios, respectively. **CUBE-LLM** also shows competitive results in general MLLM benchmarks such as refCOCO for 2D grounding with (87.0) average score, as well as visual question answering benchmarks such as VQAv2, GQA, SQA, POPE, etc. for complex reasoning.

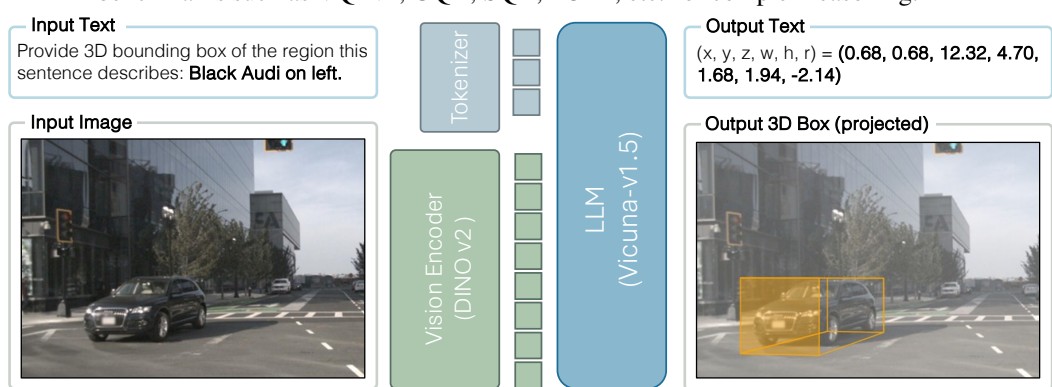

Figure 1: The overview of **CUBE-LLM** for 3D-grounding. The task requires a model to take an image, understand the input text prompt (e.g., "*Black Audi on left.*") and ground it in 3D space.

## 1 INTRODUCTION

Internet-scale visual data have brought forth the advent of multi-modal large language models (MLLMs). Rich and diverse visual supervision aligns pre-trained large language models with billions

---

*Work done during his internship at NVIDIA. Correspondence to: janghyuncho7@utexas.edu.

†Equal advising

of parameters to visual modality. The best MLLMs can recognize, understand, and reason about images and videos far better than any of specially designed architectures and algorithms (gpt; Team et al., 2023). The decades worth of computer vision datasets —image classification, captioning, object detection, grounding, document parsing, optical character recognition (OCR)— fuels the powerful MLLMs through joint training as a next token prediction task. Introducing the ability to "ground" in 2-dimensional space (*image coordinates*) bridges the low-level perception to high-level reasoning about visual input, much like human cognition. However, one critical difference is that we perceive the world in 3-dimensional space (*view coordinates*). This 3-dimensional grounding allows us to perceive and reason about the visual input closer to the actual world, which the current state of MLLMs has not explored yet.

In this work, our goal is to develop a framework to train an MLLM capable of reasoning in both 2D and 3D spaces. We demonstrate that pure data scaling can achieve our goal without any 3D-specific architectural design or training objective. We instead focus on careful data curation to address one question: *what tasks will induce 2D to 3D generalization?* To this end, we introduce a large-scale language-image pretraining dataset for 2D and 3D, called LV3D. We start by combining a diverse collection of 2D and 3D vision datasets for indoors and outdoors and standardize labels to follow the consistent format across datasets. We blend in the vision datasets with instruction-following data of MLLM training as a series of question-answer pairs (§ 3.1). Next, we augment our blended datasets by decomposing the vision labels into easier tasks (*e.g., 3D box → 2D point, depth, size, orientation*). This trains our model to adapt to versatile input and output formats and connects the underlying 2D and 3D structure (§ 3.2). Most importantly, we mix in a series of QA pairs about an object for "step-by-step" reasoning, from easier (*e.g., 2D box*) to harder (*e.g., 3D box*) tasks. This directly induces 2D to 3D generalization due to the autoregressive nature of MLLMs (§ 3.3). Finally, we train a MLLM on LV3D as a single "next token prediction" task, called **CUBE-LLM** (§ 3.4).

**CUBE-LLM** exhibits several intriguing properties. First, **CUBE-LLM** can self-improve its 3D reasoning performance by prompting with its own 2D predictions. This *visual chain-of-thought reasoning* resembles the well-known behavior of LLMs (Wei et al., 2022b). Second, **CUBE-LLM** can adapt to versatile input and output formats and questions, which follows *instruction following* ability of LLMs (Wei et al., 2022a). Finally, **CUBE-LLM** can be *prompted* with any specialist models for any additional modalities (*e.g., LiDAR*) by simply adding their predictions to the question. **CUBE-LLM** shows remarkable improvement with data-scaling in both 2D and 3D, for indoor and outdoor scene grounding as well as complex reasoning tasks such as QA in driving scenarios.

We evaluate our model's performance in both 3D grounding and 3D complex reasoning tasks on various indoor and outdoor datasets as well as a standard MLLM benchmark and show qualitative results in 3D grounding in non-driving scenes (Fig. 2). For 3D grounding on the Talk2Car dataset (Deruyttere et al., 2019), **CUBE-LLM** surpasses the baselines by **21.3** in Bird's Eye View (BEV) AP (71.4 vs 50.1) and by **18.7** in 3D AP (64.1 vs 45.4). Additionally, our training framework improves the performance of **CUBE-LLM** on the DriveLM (Sima et al., 2023) dataset, nearly doubling the performance in the BEV AP (66.0 vs 33.2) for 3D grounding from a baseline. We also test **CUBE-LLM** on complex reasoning benchmark of driving scenarios (DriveLM), and improve the overall score by **17.7** (50.1 vs 32.4) compared to DriveLM baseline (Sima et al., 2023). Furthermore, we show that **CUBE-LLM** performs the state-of-the-art in 2D referring expression comprehension, achieving the average score of **87.0** on refCOCO/+/g. Finally, we show that **CUBE-LLM** maintains competitive performance in various MLLM benchmarks including VQAv2, GQA, etc., confirming that our 3D reasoning capability is an *expansion*, not a *trade-off*.

## 2 RELATED WORK

**Vision Language Models.** By scaling up pre-training on the internet-scale dataset, there has been significant progress for VLMs in the 2D vision-language domain, showing strong capabilities in few-shot generalization. VLBRRT (Su et al., 2020) and ViLBERT (Lu et al., 2019) capitalized on a BERT-style framework for image-text co-embedding. CLIP (Radford et al., 2021) embedded images and text captions into a shared feature space via contrastive learning and pioneered zero-shot vision tasks. BLIP (Li et al., 2022) and BLIP2 (Li et al., 2023a) further improved CLIP by leveraging extra pseudo-labeled data and better image/language encoders. Flamingo (Alayrac et al., 2022) and its open-source implementation Open-Flamingo (Awadalla et al., 2023) proposed a fast adaptation

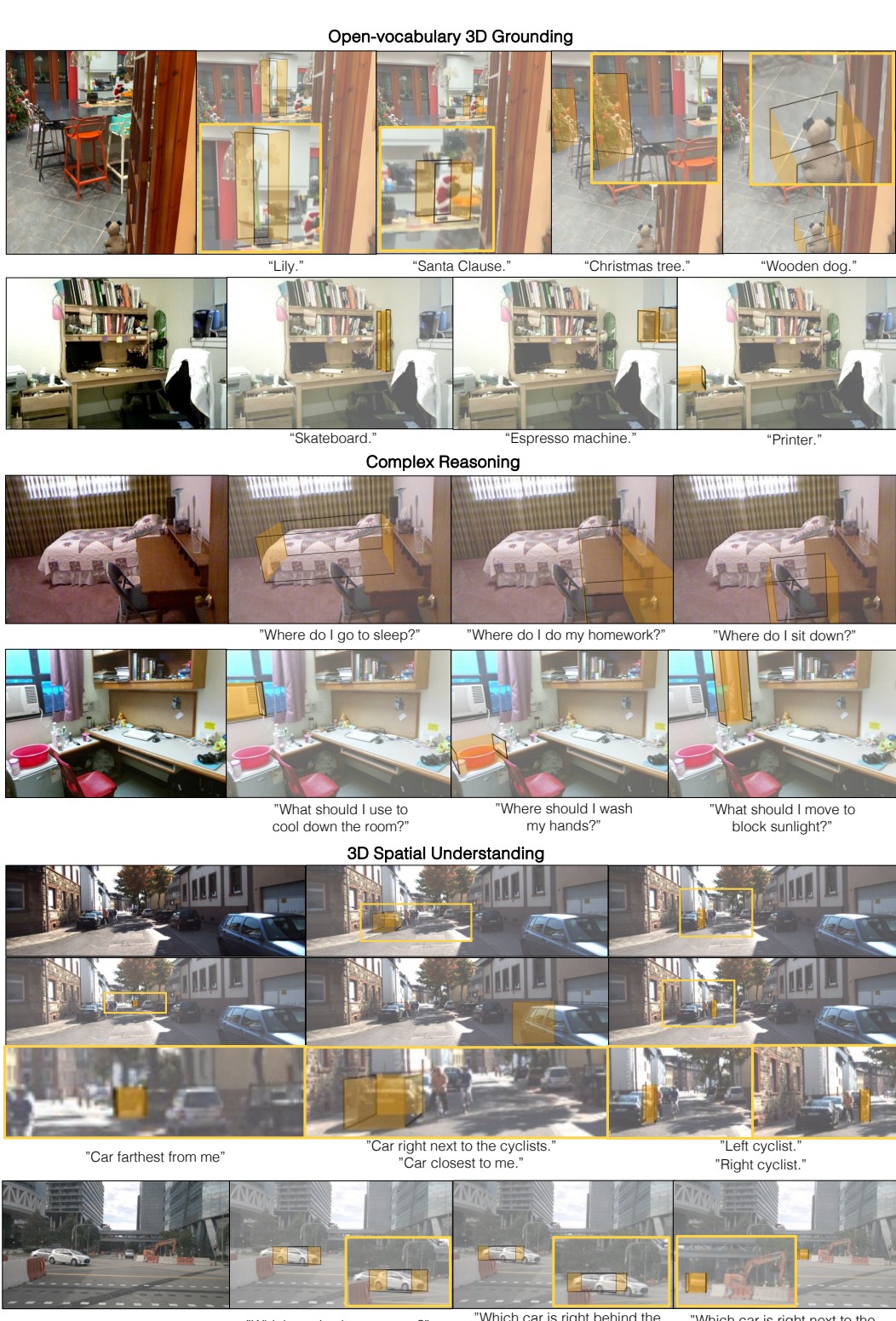

Figure 2: **Qualitative results** of CUBE-LLM 3D grounding: *open-vocabulary understanding* (**top**), *complex reasoning* (**middle**), and *3D spatial understanding* (**bottom**). Best viewed in color, **zoomed**.

approach to enable in-context few-shot learning on novel visual-language tasks. GPT4V (gpt) and Gemini (Team et al., 2023) further demonstrated state-of-the-art human-level visual reasoning ability through scaling. LLaVA (Liu et al., 2023b) pioneered instruction fine-tuning in the multimodal

field. These works have predominantly focused on 2D vision and language tasks. On the other hand, we aim to adapt these MLLMs to enhance their capabilities for complex 3D reasoning and scene understanding tasks.

**Image-grounded Reasoning.** With the advancement of multi-modal large language models, image-grounded reasoning (referring and grounding) has shown great progress in 2D space. Image-grounded reasoning requires a model to localize an object or a region that an input prompt enquires or describes about a region of interest. VisionLLM (Wang et al., 2024b) adapts a 2D object detector to align with an LLM, and GPT4-ROI (Zhang et al., 2023b) employs hierarchical feature modeling of detectors to reason about input visual prompt (ROI). Kosmos-2 (Peng et al., 2023) and Shikra (Chen et al., 2023b) have shown pure transformer-based visual encoder can surpass using 2D detectors with data scaling. Recently, Ferret (You et al., 2023) has shown remarkable image-grounded reasoning from both free-form visual prompts and text prompts. In addition, Set-of-Mark (Yang et al., 2023) shows using visual marks on image from specialists allows frontier MLLM (gpt) to do image-grounded reasoning well. These works reason in 2D space (image coordinate). To the best of our knowledge, our work is the first to expand the reasoning capability of a MLLM to 3-dimensional space.

## 3 UNIFIED LANGUAGE-IMAGE PRETRAINING FOR 2D AND 3D

Our goal is to expand the capabilities of vision-language models to reason in 3-dimensional space. We propose a unified training framework to learn from both 2D and 3D perceptual data as well as standard image-text pairs. In this section, we first discuss the data standardization to train a vision-language model at scale (Sec. 3.1), task scaling to understand perceptual information in versatile I/O format (Sec. 3.2), *visual chain-of-thought* reasoning for 3D grounding and question answering tasks (Sec. 3.3), and finally, we present **CUBE-LLM**, the final model of our unified training framework built on LLaVA-1.5 (Liu et al., 2023a) (Sec. 3.4).

### 3.1 DATA-SCALING FOR IMAGE-BASED 3D REASONING

Our goal is to train a single 2D + 3D MLLM from all data sources available. To standardize many different 2D and 3D grounding tasks into one, we standardize the data, phrase all tasks as next token prediction, and format 3D reasoning as a multi-turn conversation.

**Data standardization.** We consider points and boxes as our main spatial representation for 2D and 3D reasoning. We convert every label to either a point $o_{\text{point}}^{\text{2D}} = [\hat{x}, \hat{y}]$ or a bounding box $o_{\text{box}}^{\text{2D}} = [\hat{x}, \hat{y}, \hat{x}', \hat{y}']$. Similarly, we convert every 3D label to either $o_{\text{point}}^{\text{3D}} = [x, y, z]$ or $o_{\text{box}}^{\text{3D}} = [x, y, z, w, h, l, r_1, r_2, r_3]$ where $r_1$, $r_2$, $r_3$ are Euler angles. We first standardize image-based 3D datasets by unifying camera parameters. We follow the procedure of Omni3D (Brazil et al., 2023); define a virtual camera with a fixed focal length $f$ and transform depth $z$ according to the original camera parameters and the target image size. Since all 3D labels are unified to a consistent camera intrinsic, we can now convert all x and y coordinates to 2D projected coordinates $(\hat{x}, \hat{y})$. Consequently, we can represent all label formats to naturally follow 2D to 3D per-object token sequence:

$$o_{\text{point}}^{\mathbf{2D}} = [\hat{x}, \hat{y}] \tag{1}$$

$$o_{\text{box}}^{\mathbf{2D}} = [\hat{x}, \hat{y}, \hat{x}', \hat{y}'] \tag{2}$$

$$o_{\text{point}}^{\mathbf{3D}} = [\hat{x}, \hat{y}, z] \tag{3}$$

$$o_{\text{box}}^{\mathbf{3D}} = [\hat{x}, \hat{y}, z, w, h, l, r_1, r_2, r_3] \tag{4}$$

where each value is represented as a short sequence of text tokens (3 for 3-decimal numbers). This allows the model to predict consistent ordering of token sequence from 2D to 3D, which improves the understanding of the underlying structure. With autoregressive models, we first localize objects in image coordinates $(\hat{x}, \hat{y})$, then infer depth $(z)$, and then infer the size and orientation $(w, h, l, r_1, r_2, r_3)$.

**3D reasoning as multi-turn conversations.** Now, we combine the 2D and 3D data with language-image instruction tuning data of visual language models (Liu et al., 2023b). For each image and a set of object labels pair, we construct a multi-turn conversational question-answer data ($\mathbf{Q}_1$, $\mathbf{A}_1$, $\mathbf{Q}_2$, $\mathbf{A}_2$,

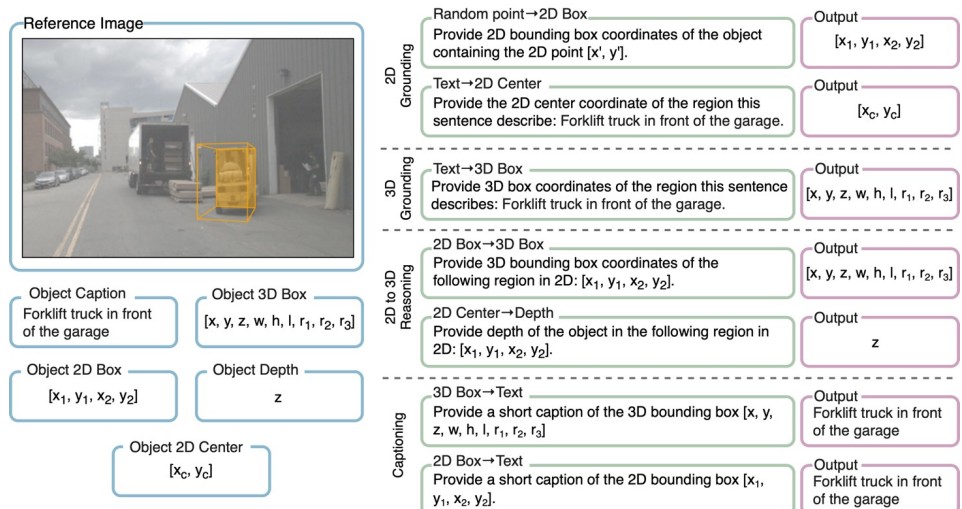

Figure 3: **Task-scaling for versatile I/O format.** Decomposing the existing label formats for 3D grounding task. A complete 3D location can be decomposed into a center point (`[x, y, z]`), a depth (`[z]`), a (projected) 2D point (`[x_c, y_c]`), and a (projected) 2D box (`[x1, y1, x2, y2]`). We define various tasks that connect among these to train versatile I/O formats. **Left**: available (decomposed) annotations. **Right**: various tasks for training.

$\ldots, \mathbf{Q}_n, \mathbf{A}_n)$. Each question refers an object with one property $b_q$ and enquires $b_a$:

$$b_q, b_a \in \{\text{box}_{\text{2D}}, \text{caption}, \text{box}_{\text{3D}}\} \tag{5}$$

Each object property has a set of prompts predefined, such as "`Provide the 3D bounding box of the region this sentence describes: <caption>`" for $b_q =$ caption and $b_a = \text{box}_{\text{3D}}$. We combine the meta information of objects (e.g., attribute, physical state, etc.) with the class name to enrich the textual information.

## 3.2 TASK-SCALING FOR VERSATILE I/O FORMAT

We are interested in a generalist model that accepts input and generates output in versatile formats. Users may want to supplement 2D points or boxes as visual prompts during inference, or may only want the metric depth of an object instead of a complete 3D location. This interest in versatile I/O format shares the same spirit of instruction tuning in 2D-based visual language models (Liu et al., 2023b; Dai et al., 2023; Alayrac et al., 2022). To this end, we define multiple relevant tasks for a model to adapt to a wider spectrum of similar tasks in 2D and 3D. We start by *decomposing* the existing label formats to easier tasks as illustrated in Figure 3. After, we have expanded the set of object properties to construct question-answer pairs:

$$b_q \in \{\text{point}_{\text{2D}}, \text{box}_{\text{2D}}, \text{caption}, \text{point}_{\text{3D}}, \text{box}_{\text{3D}}\} \tag{6}$$
$$b_a \in \{\text{point}_{\text{2D}}, \text{box}_{\text{2D}}, \text{caption}, \text{depth}, \text{point}_{\text{3D}}, \text{box}_{\text{3D}}\} \tag{7}$$

We construct up to $n = 30$ question answer pairs $(\mathbf{Q}_{b_a}^{b_q}, \mathbf{A}_{b_a})$ sampled at random for each data. We combine a collection of 2D and 3D vision datasets (LV3D), summarized in Table 1, and jointly train with this expanded set of tasks.

## 3.3 VISUAL CHAIN-OF-THOUGHT PROMPTING

One of the most intriguing properties of large language models is its *emergent* ability to improve reasoning with intermediate steps (Wei et al., 2022b). This mostly attributes to a vast corpus of rich text data with numerous step-by-step question-answering samples (Wei et al., 2022a). We artificially supplement this *step-by-step* reasoning of 3D by interleaving multiple questions of the same object from easy-to-hard order (the left part of Figure. 4):

$$\text{maximize} \quad \begin{cases} p(\mathbf{A}_{\text{box}_{\text{2D}}} | \mathbf{Q}_{\text{box}_{\text{2D}}}^{\text{caption}}) & \text{question 1} \\ p(\mathbf{A}_{\text{box}_{\text{3D}}} | \mathbf{Q}_{\text{box}_{\text{2D}}}^{\text{caption}}, \mathbf{A}_{\text{box}_{\text{2D}}}, \mathbf{Q}_{\text{box}_{\text{3D}}}^{\text{caption}}) & \text{question 2} \\ \ldots \end{cases} \tag{8}$$

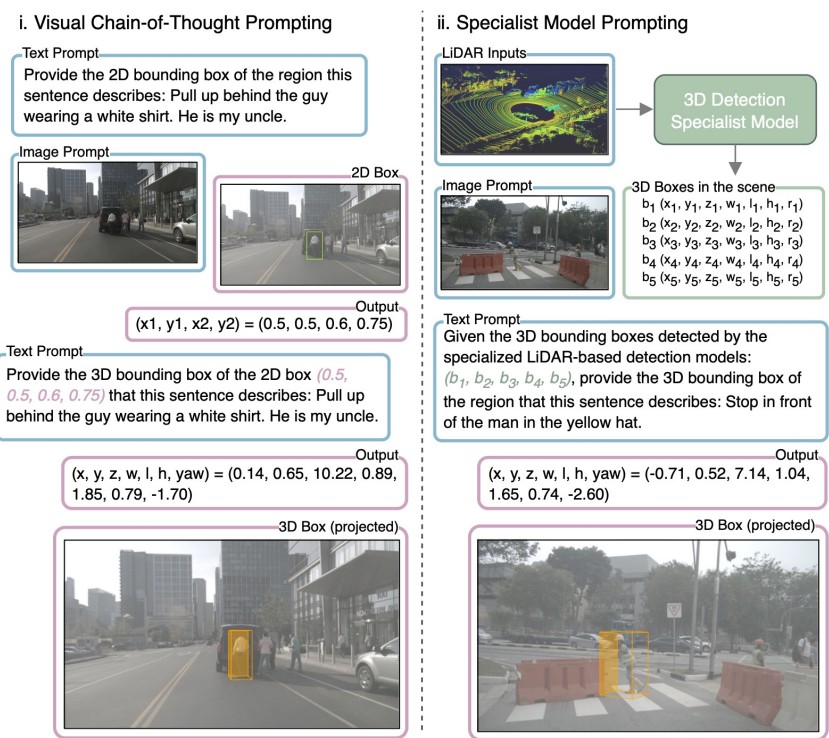

Figure 4: **CUBE-LLM inference with prompting**. **Left**: Visual Chain-of-Thought Prompting to reason in 3D step-by-step. **Right**: Incorporating specialist models to further improve localization of **CUBE-LLM**. Our model can either predict directly from text prompts, with visual chain-of-thought prompting, or with specialist predictions as prompts. Figure 9 and 10 in appendix visualize these.

Furthermore, we allow test-time adaptation to any specialist models by mixing in *candidate* objects as a system prompt (the right part of Figure. 4). This effectively alleviates the problem of localizing in 3D to "choosing the appropriate box from candidates",

$$\text{maximize} \quad p(\mathbf{A}_{\text{box3D}}|\mathbf{S}_{\text{box3D}}, \mathbf{Q}_{\text{box3D}}^{\text{caption}}) \quad (9)$$

where $\mathbf{S}_{\text{box3D}}$ is a set of candidate boxes, which can be provided by any specialist models (depending on available input modalities) at inference. During training, we use the ground truth boxes with a prompt "Here is the list of 3D bounding boxes of all objects around the camera:" and our model does not bind with any particular specialist model.

### 3.4 CUBE-LLM

We introduce **CUBE-LLM**, a multi-modal large language model based on LLaVA-1.5 architecture trained to reason in both 2D and 3D. Although we maintain the generality of model architecture, we make simple yet critical changes to the original LLaVA. We first replace the CLIP visual encoder with DINOv2 (Oquab et al., 2024) and undergo the same alignment step of the original LLaVA. Although DINOv2 is not a text-aligned visual encoder like CLIP, we found minimal degradation in the standard visual language model benchmarks while significantly improving 3D-related tasks. Then, we finetune the language model (Vicuna-7B (Chiang et al., 2023)) while freezing the visual encoder and jointly on LLaVA instruction-following data and the 2D part of LV3D following Sec. 3.1, 3.2 and 3.3. We use low image resolution ($336 \times 336$) and train with a large batch size. Then, we proceed an additional finetuning stage for both visual and language models with high-resolution images ($672 \times 672$) of the full LV3D. More details can be found in the section A and Figure 8 of the appendix.

## 4 EXPERIMENTS

We evaluate **CUBE-LLM** in three aspects: (1) 3D-grounded reasoning for indoor and outdoor scenes, (2) complex reasoning in 3D, and (3) standard MLLM benchmarks such as 2D grounding and VQA.

Table 1: **2D and 3D Language-Image Pretraining Dataset (LV3D)**. Summary of components detailing the number of images, tasks, availability of 2D and 3D labels, the number of QAs and objects, and their multiples during training (*stage 1* and *stage 2*). ⋆: Only used 2D bounding box.

| dataset | images | labels$_{2D}$ | labels$_{3D}$ | captions | # QAs | stage 1 | stage 2 |
|---|---|---|---|---|---|---|---|
| LLaVA data (Liu et al., 2023b) | 80K | ✓ | ✗ | ✓ | 158K | 1 | 0.5 |
| refCOCO/+/g (Yu et al., 2016) | 67K | ✓ | ✗ | ✓ | 154K | 1 | 0.5 |
| GRIT (subset) (Peng et al., 2023) | 4M | ✓ | ✗ | ✓ | 6.9M | 1 | 0.3 |
| AS (filtered) (Wang et al., 2024a) | 3.7M | ✓ | ✗ | ✓ | 13.2M | 1 | 0.5 |
| COCO (Lin et al., 2014) | 118K | ✓ | ✗ | ✗ | 860K | 1 | 0.5 |
| Objects365 (Shao et al., 2019) | 600K | ✓ | ✗ | ✗ | 25.4M | 0.3 | 0.2 |
| SUN-RGBD (Song et al., 2015) | 5K | ✓ | ✓ | ✗ | 41K | 1⋆ | 5 |
| Hypersim (Roberts et al., 2021) | 67K | ✓ | ✓ | ✗ | 2M | 1⋆ | 5 |
| ArkitScenes (Baruch et al., 2021) | 53K | ✓ | ✓ | ✗ | 420K | 1⋆ | 5 |
| Objectron (Ahmadyan et al., 2021) | 37K | ✓ | ✓ | ✗ | 43K | 1⋆ | 5 |
| KITTI (Geiger et al., 2012) | 4K | ✓ | ✓ | ✗ | 25K | 1⋆ | 5 |
| NuScenes (Caesar et al., 2019) | 40K | ✓ | ✓ | ✗ | 1.1M | 1⋆ | 2 |
| Lyft (Houston et al., 2021) | 105K | ✓ | ✓ | ✗ | 723K | 0 | 2 |
| Argoverse2 (Wilson et al., 2021) | 79K | ✓ | ✓ | ✗ | 915K | 0 | 4 |
| Waymo (Sun et al., 2020) | 680K | ✓ | ✓ | ✗ | 5.1M | 0 | 0.4 |
| Total | 9.6M | ✓ | ✓ | ✓ | 40.9M | 0.87 | 0.52 |

## 4.1 DATASETS

We pre-train **CUBE-LLM** on LV3D, a large collection of 2D and 3D dataset (Table 1). We format the existing labels into multi-turn instruction-following tasks under standard format, as described in Section 3.1, 3.2, and 3.3. We describe details of dataset construction in the section C of the appendix. We evaluate our model on diverse tasks, including the following 3D grounding datasets.

**Talk2Car** (Deruyttere et al., 2019) is a 3D referring expression comprehension dataset of various driving scenarios. It consists of 8,349 training samples and 1,163 validation samples with images and LiDAR data. It provides rich question-answer pairs grounded to an object in the image. Each object is labeled with a situational text that uniquely identifies the object (e.g., "*Wow hold on! That looks like my stolen bike over there! Drop me off next to it.*"). The original benchmark (Deruyttere et al., 2019) evaluates the 2D grounding performance with the AP$_{0.5}$ metric. MSSG (Cheng et al., 2023) extends the task to 3D grounding and evaluates on both BEV AP and 3D AP.

**DriveLM** (Sima et al., 2023) is a recently released question-answering dataset for driving scenarios based on NuScenes (Caesar et al., 2019). It consists of multi-view images and LiDAR point clouds as well as frame-level QA data, total of 4,871 frames. Each frame covers core AV tasks such as perception, prediction, and planning, as well as a scene description and 2D boxes of important objects. We process DriveLM and construct a 3D grounding dataset, which we call **DriveLM-Grounding**. We evaluate 3D grounding with the same BEV AP and 3D AP metric as those in Talk2Car. We also use the original **DriveLM-QA** data to fine-tune **CUBE-LLM** for complex reasoning tasks. We sample 600 scenes for training and 96 scenes for validation, which we include the DriveLM provided scenes for sample evaluation and Talk2Car validation split scenes.

The remaining details of the evaluation datasets will be in the section C of the appendix.

## 4.2 3D-GROUNDED REASONING

In Table 5, we show 2D and 3D grounding results of **CUBE-LLM** and baselines on Talk2Car dataset. The baselines that rely solely on camera inputs are only evaluated on 2D grounding, whereas those incorporating both camera and LiDAR inputs are evaluated on both 2D and 3D grounding. **CUBE-LLM** is pre-trained on LV3D and fine-tuned on Talk2Car with resolution 672 × 672. We apply a visual chain of thought when predicting the 3D grounding. Remarkably, our camera-only **CUBE-LLM** significantly surpasses the state-of-the-art model FA (Deruyttere et al., 2022) by 5.7 points on 2D AP$_{0.5}$. Surprisingly, **CUBE-LLM** also outperforms the camera+LiDAR baseline, Talk2Car-3D (Deruyttere et al., 2019), by 15.7 points on the BEV AP$_A$ metric (Cheng et al., 2023). Our camera-only **CUBE-LLM** is only 3.8 points behind the state-of-the-art camera+LiDAR baseline MSSG (Cheng et al., 2023). MSSG (Cheng et al., 2023) utilized the LiDAR point encoder similar to CenterPoint (Yin et al., 2021) as well as image and text encoders for predicting 3D

Figure 5: **Talk2Car Benchmark for 2D and 3D Grounding**. We denote C as Camera and L as LiDAR. †: we use the top-30 predicted boxes of CenterPoint (Yin et al., 2021) as visual prompt as illustrated in Figure 4. $AP_A$ and $AP_B$ follow MSSG (Cheng et al., 2023) that apply different IoU threshold for each category. **Top**: Zeroshot Talk2Car result with varying LV3D data scale in %. **Bottom**: Zeroshot Talk2Car result with and without V-CoT training samples (Sec. 3.3) and 2D → 3D stage training (Sec. 3.4)

| Method | Input | 2D $AP_{0.5}$ | BEV $AP_A$ | $AP_B$ | 3D $AP_A$ | $AP_B$ |
|---|---|---|---|---|---|---|
| *2D Specialist* | | | | | | |
| Talk2Car-2D (Deruyttere et al., 2019) | C | 50.5 | - | - | - | - |
| VL-Bert (Su et al., 2020) | C | 63.1 | - | - | - | - |
| ViLBERT (Lu et al., 2019) | C | 68.9 | - | - | - | - |
| CMRT (Luo et al., 2020) | C | 69.1 | - | - | - | - |
| Stacked VLBert (Dai et al., 2020) | C | 71.0 | - | - | - | - |
| FA (Deruyttere et al., 2022) | C | 73.5 | - | - | - | - |
| **CUBE-LLM** [7b] (zero-shot) | C | 49.2 | 32.0 | 19.5 | 22.3 | 9.8 |
| **CUBE-LLM** [7b] | C | **79.2** | 46.3 | 30.1 | 34.7 | 18.2 |
| **CUBE-LLM** [13b] (zero-shot) | C | 54.9 | 35.9 | 23.6 | 26.1 | 10.7 |
| *3D Specialist* | | | | | | |
| Talk2Car-3D (Deruyttere et al., 2019) | L + C | - | 30.6 | 24.4 | 27.9 | 19.1 |
| MSSG (Cheng et al., 2023) | L + C | - | 50.1 | 35.7 | 45.4 | 23.7 |
| **CUBE-LLM** [7b,†] | L + C | 76.3 | **71.4** | **61.2** | **64.1** | **39.8** |

Zeroshot Talk2Car

Table 2: **DriveLM QA and Grounding Benchmarks**. (**Left**) †: finetuned LLaVA-1.5. DriveLM baseline based on LLaMA Adapter V2 (Gao et al., 2023). **Top**: same split as the DriveLM baseline. **Bottom**: our larger test split held-out from all training. ‡: reported DriveLM result on the full test set. (**Right**) LV3D (2D) indicates that only 2D data in the pre-train dataset is included. We finetune **CUBE-LLM** and LLaVA-1.5 (Liu et al., 2023a) on the DriveLM-Grounding dataset.

(a) **DriveLM-QA**

| Method | Acc. | Match | Overall |
|---|---|---|---|
| *baseline split* | | | |
| DriveLM baseline | 0.0 | 28.3 | 32.4 |
| LLaVA-1.5† | 38.5 | 26.1 | 36.1 |
| **CUBE-LLM** | **38.5** | **39.0** | **50.1** |
| *our split* | | | |
| DriveLM baseline‡ | 0.0 | 18.8 | 32.8 |
| LLaVA-1.5† | 24.1 | 36.4 | 43.8 |
| **CUBE-LLM** | **32.4** | **39.2** | **45.4** |

(b) **DriveLM-Grounding**

| Method | $AP^{BEV}_A$ | $AP^{BEV}_B$ | $AP^{3D}_A$ | $AP^{3D}_B$ |
|---|---|---|---|---|
| *finetune* | | | | |
| LLaVA-1.5 | 33.2 | 16.3 | 21.7 | 7.7 |
| CLIP → DINOv2 | 39.6 | 21.7 | 25.8 | 10.5 |
| + LV3D (2D) | 50.5 | 31.2 | 32.5 | 17.3 |
| + LV3D (3D) | **66.0** | **52.1** | **56.2** | **40.5** |

grounding. Similarly, we leverage the LiDAR modality by using the top-30 predictions from CenterPoint (Yin et al., 2021) as input prompt of **CUBE-LLM**. We observe a substantial 25.1 points improvement in $AP_A$, outperforming MSSG (Cheng et al., 2023) by 21.3 points. Furthermore, we observe a similar trend on the DriveLM-Grounding dataset, shown in Table 2. **CUBE-LLM** shows significant improvements compared to directly finetuning from LLaVA-1.5, resulting in a 32.8 points improvement on the BEV $AP_A$ metric. Lastly, we show indoor 3D grounding in Table 3, where we compare **CUBE-LLM** trained with *LV3D-small* and LV3D. LV3D-small contains the same indoor 3D dataset but without the most of outdoor data. Under our training framework, outdoor data scaling translates to indoor well. We describe the detailed experiment setting in the section C of the appendix.

**Ablations.** In Figure 5 (right), we first show **CUBE-LLM** exhibits an impressive scalability in 3D grounding task. Next, we show that employing the visual chain-of-thought samples during training improves zeroshot 3D AP by 3.2 points. The process of V-CoT and Specialist Promptings are illustrated in Figure 6 or in Figure 9 and 10 in the appendix.

## 4.3 COMPLEX REASONING IN 3D

To show the effectiveness of 3D reasoning capability, we finetune **CUBE-LLM** on DriveLM-QA dataset (Table 2). We compare **CUBE-LLM** with LLaVA-1.5 (Liu et al., 2023a) to show the impact of our pretraining, as well as the official baseline (Sima et al., 2023). All models use 7-B scale LLM (Vicuna-7B (Chiang et al., 2023) or LLaMA-7B (Touvron et al., 2023)) and are fine-tuned on a subset

Table 3: **Indoor 3D Grounding Benchmark.** Here we compare **CUBE-LLM** trained on "small" subset of LV3D and the full LV3D. Although the subset and full LV3D share the same indoor datasets, the added 2D data and outdoor 3D data translate to better indoor 3D grounding result.

| Pre-train Data | Objectron | | ArkitScenes | | SUN-RGBD | |
|---|---|---|---|---|---|---|
| | $\text{mAP}_{3D}^{cls}$ | $\text{mAP}_{3D}^{cls+loc}$ | $\text{mAP}_{3D}^{cls}$ | $\text{mAP}_{3D}^{cls+loc}$ | $\text{mAP}_{3D}^{cls}$ | $\text{mAP}_{3D}^{cls+loc}$ |
| LV3D-small | 56.7 | 36.1 | 21.6 | 28.3 | 25.5 | 25.5 |
| LV3D | 69.8 | 45.4 | 23.5 | 31.8 | 29.7 | 28.8 |
| Δ | **13.1** | **9.3** | **1.9** | **3.5** | **4.2** | **3.3** |

Table 4: **Referring Expression Comprehension Benchmark**. We compare **CUBE-LLM** with other MLLMs for general 2D grounding tasks. **CUBE-LLM** consistently performs best in all data splits.

| Models | Size | RefCOCO | | | RefCOCO+ | | | RefCOCOg | | Avg. |
|---|---|---|---|---|---|---|---|---|---|---|
| | | val | testA | testB | val | testA | testB | val | test | |
| *Specialist* | | | | | | | | | | |
| MAttNet (Yu et al., 2018) | | 76.4 | 80.4 | 69.3 | 64.9 | 70.3 | 56.0 | 66.7 | 67.0 | 68.9 |
| OFA-L (Wang et al., 2022) | | 80.0 | 83.7 | 76.4 | 68.3 | 76.0 | 61.8 | 67.6 | 67.6 | 72.7 |
| TransVG (Deng et al., 2021) | | 81.0 | 82.7 | 78.4 | 64.8 | 70.7 | 56.9 | 68.7 | 67.7 | 71.4 |
| UNITER (Chen et al., 2020b) | | 81.4 | 87.0 | 74.2 | 75.9 | 81.5 | 66.7 | 74.0 | 68.7 | 76.2 |
| VILLA (Gan et al., 2020) | | 82.4 | 87.5 | 74.8 | 76.2 | 81.5 | 66.8 | 76.2 | 76.7 | 77.8 |
| UniTAB (Yang et al., 2022) | | 86.3 | 88.8 | 80.6 | 78.7 | 83.2 | 69.5 | 80.0 | 80.0 | 80.6 |
| MDETR (Kamath et al., 2021) | | 86.8 | 89.6 | 81.4 | 79.5 | 84.1 | 70.6 | 81.6 | 80.9 | 81.8 |
| Grounding DINO L (Liu et al., 2023c) | | 90.6 | 93.2 | 88.2 | 82.8 | 89.0 | 75.9 | 86.1 | 87.0 | 86.6 |
| *Generalist* | | | | | | | | | | |
| LLaVA-1.5 (Liu et al., 2023a) | 7B | 75.6 | 82.1 | 66.9 | 65.5 | 76.2 | 53.9 | 68.9 | 69.1 | 69.8 |
| VisionLLM-H (Wang et al., 2024b) | 7B | 86.7 | - | - | - | - | - | - | - | - |
| Shikra (Chen et al., 2023b) | 7B | 87.0 | 90.6 | 80.2 | 81.6 | 87.4 | 72.1 | 82.3 | 82.2 | 82.9 |
| Ferret (You et al., 2023) | 7B | 87.5 | 91.4 | 82.5 | 80.8 | 87.4 | 73.1 | 83.9 | 84.8 | 83.9 |
| MiniGPT-v2 (Chen et al., 2023a) | 7B | 88.7 | 91.7 | 85.3 | 80.0 | 85.1 | 74.5 | 84.4 | 84.7 | 83.8 |
| LLaVA-G (Zhang et al., 2023a) | 7B | 89.2 | - | - | 81.7 | - | - | 84.8 | - | - |
| Qwen-VL (Bai et al., 2023) | 7B | 88.6 | 92.3 | 84.5 | 82.8 | 88.6 | 76.8 | 86.0 | 86.3 | 85.7 |
| **CUBE-LLM** | 7B | **90.9** | **92.6** | **87.9** | **83.9** | **89.2** | **77.4** | **86.6** | **87.2** | **87.0** |
| Shikira (Chen et al., 2023b) | 13B | 87.8 | 91.1 | 81.8 | 82.9 | 87.8 | 74.4 | 82.6 | 83.2 | 84.0 |
| Ferret (You et al., 2023) | 13B | 89.5 | 92.4 | 84.4 | 82.8 | 88.1 | 75.2 | 85.8 | 86.3 | 85.6 |
| **CUBE-LLM** | 13B | **91.8** | **93.5** | **88.6** | **86.0** | **90.8** | **79.1** | **87.6** | **88.6** | **88.3** |

Table 5: **MLLM Benchmarks**. We compare **CUBE-LLM** in various general MLLM tasks.

| Model | Size | $\text{VQA}^{v2}$ | GQA | VizWiz | $\text{SQA}^{I}$ | POPE |
|---|---|---|---|---|---|---|
| BLIP-2 (Li et al., 2023a) | 13B | 41.0 | 41.0 | 19.6 | 61.0 | 85.3 |
| InstructBLIP (Dai et al., 2023) | 7B | - | 49.2 | 34.5 | 60.5 | - |
| InstructBLIP (Dai et al., 2023) | 13B | - | 49.5 | 33.4 | 63.1 | 78.9 |
| IDEFICS (Laurençon et al., 2023) | 9B | 50.9 | 38.4 | 35.5 | - | - |
| Shikra (Chen et al., 2023b) | 13B | 77.4 | - | - | - | - |
| Qwen-VL (Bai et al., 2023) | 7B | 78.8 | 59.3 | 35.2 | 67.1 | - |
| Qwen-VL (chat) (Bai et al., 2023) | 7B | 78.2 | 57.5 | 38.9 | 68.2 | - |
| miniGPT-v2 (Chen et al., 2023a) | 7B | - | 60.1 | **53.6** | - | - |
| LLaVA-1.5 (Liu et al., 2023a) | 7B | 78.5 | 62.0 | 50.0 | 66.8 | 85.9 |
| LLaVA-1.5 (Liu et al., 2023a) | 13B | **80.0** | 63.3 | 53.6 | 71.6 | 85.9 |
| **CUBE-LLM** | 7B | 78.3 | 62.4 | 51.0 | 69.2 | 87.1 |
| **CUBE-LLM** | 13B | 79.9 | **64.1** | 53.0 | **72.2** | **88.2** |

of DriveLM train split. The top rows are the result of scenes held out by the authors and the bottom rows are our additional split to evaluate models on a larger test set. The evaluation metric is based on accuracy, match (localization), BLEU/ROUGE$_L$/CIDEr, and ChatGPT score for favorable text generation. In Figure 7, we visualize **CUBE-LLM**'s prediction for complex reasoning in driving.

## 4.4 GENERAL MLLM BENCHMARKS

We show the performance of **CUBE-LLM** on general MLLM benchmarks. In Table 4, we compare **CUBE-LLM** to the state-of-the-arts in Referring Expression Comprehension (REC) benchmark on refCOCO/+/g (Yu et al., 2016) dataset. We compare **CUBE-LLM** to *specialist* models such as MDETR (Kamath et al., 2021) and UniTAB (Yang et al., 2022) which employs detection-specific architecture, and *generalist* models of same size such as Ferret (You et al., 2023), Qwen-VL (Bai et al.,

V-CoT Prompting                                    Specialist Prompting

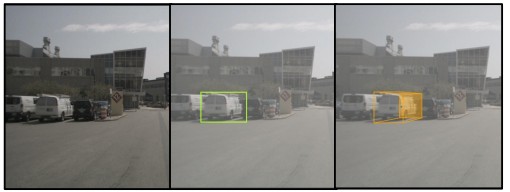
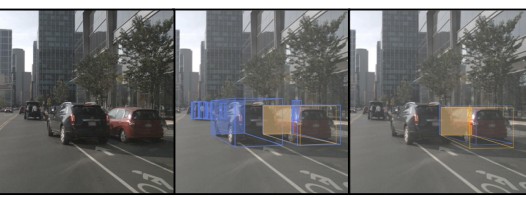

"Turn left, park behind the white van, left of the black car."          "Park behind this red car."

Figure 6: **CUBE-LLM inference with prompting**. **Left**: Visual Chain-of-Thought Prompting to reason in 3D step-by-step. **Right**: Incorporating specialist models to further improve localization of **CUBE-LLM**. Blue 3D boxes are the predictions of CenterPoint on corresponding LiDAR points.

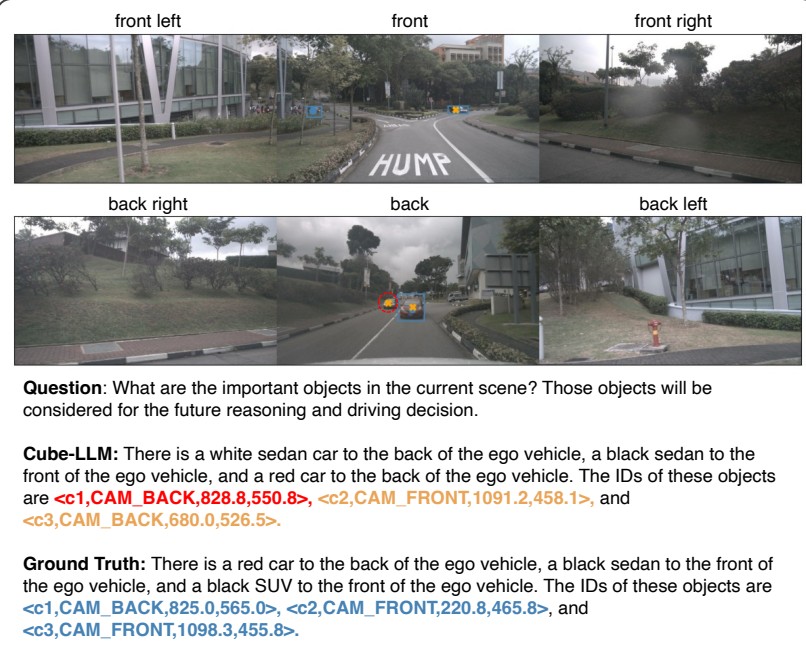

Figure 7: **CUBE-LLM prediction on DriveLM-QA**. Orange marks are predicted 2D points by **CUBE-LLM**. Blue marks are the reference marks and the corresponding bounding box in the ground truth answers. Red circle is the predicted object that does not agree with the ground truth.

2023) and MiniGPT-v2 (Chen et al., 2023a). In all test splits, **CUBE-LLM** consistently outperforms with average score of **87.0**. In Table 5, we compare **CUBE-LLM** with other competitive MLLMs of same model size on VQAv2 (Goyal et al., 2017), GQA (Hudson & Manning, 2019), VizWiz (Gurari et al., 2018), ScienceQA-Image (Lu et al., 2022), and POPE (Li et al., 2023b). The first row has models with fully zero-shot evaluation, and the bottom rows have models that have seen images from some of the datasets. Compared to LLaVA-1.5 (Liu et al., 2023a), miniGPT-v2 (Chen et al., 2023a) and Qwen-VL (Bai et al., 2023), **CUBE-LLM** maintain the competitive result, validating that our 3D understanding does not degrade general reasoning capability of MLLM.

## 5 CONCLUSION

In this paper, we present **CUBE-LLM**, a multi-modal language model that can reason in both 2D and 3D. We provide a collection of datasets (LV3D) and a training framework to effectively scale MLLM training for 3D understanding. We evaluate **CUBE-LLM** in 2D and 3D grounded reasoning and VQA tasks, and show competitive results. We also show that **CUBE-LLM** exhibits the behaviors of LLMs such as chain-of-thought prompting or visual prompting to further improve the 3D localization of our model. Finally, we show that our model can adapt to any specialist models during inference by prompting their predictions as visual prompts. We examine that pure transformer-based MLLM with minimal inductive bias can learn about 3D understanding solely by data scaling.

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

# A    TRAINING DETAILS

In this section, we provide more training and implementation details of **CUBE-LLM**.

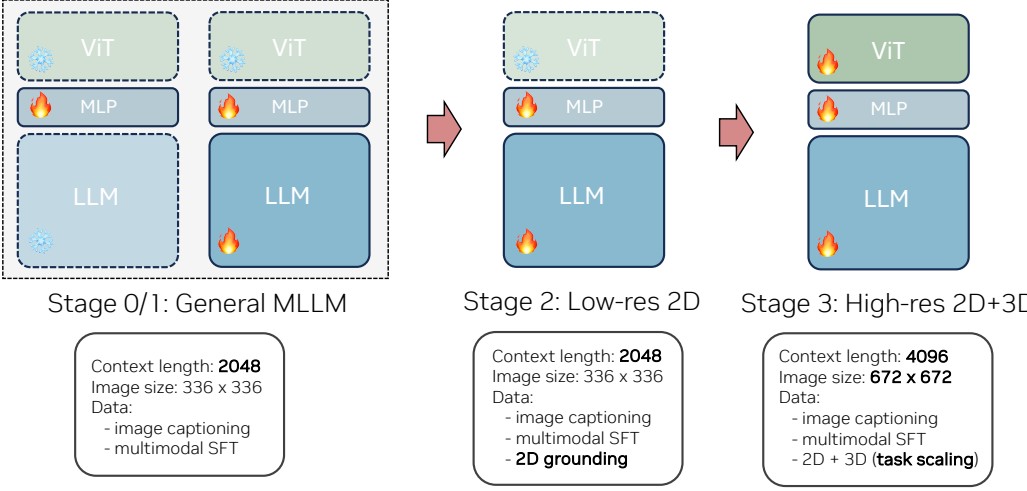

Figure 8: **Cube-LLM training pipeline.** We illustrate different stages of **CUBE-LLM** training pipeline. Stage 0 and 1 follow common MLLM training following (Liu et al., 2023a), Stage 1 trains 2D parts of LV3D in low-resolution with the vision encoder frozen. Finally, we fully finetune on all 2D and 3D parts of LV3D in high-resolution (672 × 672). In Figure 5 (bottom right, second) we compare this setup to combining the Stage 2 and 3 together.

**Implementation details.** We use LLaVA-1.5 Liu et al. (2023a) with Vicuna-7B as our base model. We replace the CLIP visual encoder with ViT-L/14 Dosovitskiy et al. (2021) based DINOv2. For all localization outputs, we use 3 decimal places with text tokens, and keep 3 tokens per value (e.g., [021,521, ...]). Accordingly, we pre-process all LLaVA instruction-following data to reflect this change. We follow the same alignment step to train the MLP projection layers with the same training setup in Liu et al. (2023a). For 2D and 3D pretraining, we use random sampling following the sampling rate in Table 1. Each data sample (image-annotation pair) is converted to the multi-turn conversation format (Fig. 3) sampled at random. During pretraining, we use 8×8 A100s with a batch size of 1024 and train the model with a learning rate $lr = 2 \times 10^{-5}$ on images with 336 × 336 resolution. Then, we fine-tune all parameters including the visual encoder on a higher resolution 672 × 672 with 8×8 A100s and a batch size of 256 with 4 gradient accumulation steps (effective batch size of 1024) and a learning rate $lr = 2 \times 10^{-5}$.

**CUBE-LLM pre-training** undergoes the pretraining stage and finetuning stage. The pretraining is done on LV3D with the dataset multiples specified in Table 1 of the main paper. In this stage, all object depth $z$ are transformed to align with the virtual camera (same practice as Omni3D Brazil et al. (2023)) and converted to log-scale. For each $(x, y, z, w, h, l, r_1, r_2, r_3)$, we normalize $x$ and $y$ in image coordinate from 0 to 999. For $z$, we set $z_{\min} = -4$ and $z_{\max} = 5$ (after log-scale) and rescale in 0 and 999. Similarly, $w_{\min} = 0, w_{\max} = 15, h_{\min} = 0, h_{\max} = 15, l_{\min} = 0, l_{\max} = 15$. All Euler angles are normalized between 0 and $2\pi$. We train all 3 Euler angles in "yaw", "pitch", and "roll" order. Such ordering of angles in pretraining ensures consistent sequential ordering before and after finetuning. To support flexible question formats during inference, we prepare a set of question templates and randomly sample one per object during training (*e.g.*, "Provide 3D bounding box of the region in the image that this sentence describes: <>" or "What is the 3D box of the <>?"). For datasets where text does not contain orientation-specific information, we apply random horizontal flip data augmentation. We shuffle object order randomly, use all objects even if there are duplicate questions, and cut off the training token sequence by the context length of the language model (4096). We pre-train with 336 × 336 image size with frozen image-encoder and 672 × 672 with full training. Figure 8 illustrates the overal training pipeline of **CUBE-LLM**. This stage-wise training more beneficial compared to fully finetuning from beginning, as compared Figure 5 (bottom right).

**CUBE-LLM fine-tuning** undergoes a few change. Since finetuning benchmarks are all for outdoor scenes, we finetune $z$ to be in meter (*i.e.,* no log-scale), and set $z_{min} = 0$, $z_{max} = 140$. We also ignore "pitch" and "roll" and only train for "yaw": $(x, y, z, w, h, l, r_1)$. We finetune on Talk2Car, DriveLM-grounding, and NuScenes dataset altogether for 10 epochs. We randomly prompt ground-truth boxes in the system prompt to allow specialist prompting at inference. We also randomly sample to query either 2D bounding box, 3D bounding box, or 2D-to-3D multi-turn question answering.

## B  ADDITIONAL RELATED WORK

**3D Scene Understanding with LLM.** There has been a great progress in multi-modal large language models that consider 3D input for scene understanding. 3D-LLM (Hong et al., 2023) processes 3D point clouds as multi-view images to extract 3D features and trains a multi-modal large language model for 3D scene understanding. Scene-LLM (Fu et al., 2024) improves this framework by introducing enhanced 3D representation and data generation. Point-LLM (Xu et al., 2024) directly takes point cloud with point encoder and finetunes a large language model for 3D object understanding and captioning tasks. These works have shown that large language models can process point cloud input and reason over it if properly trained with data. **CUBE-LLM** follows this effort but focuses on reasoning in 3D from RGB images only.

**3D Object Grounding.** There has been many works for 3D object grounding primarily with point cloud input. ScanRefer (Chen et al., 2020a) introduces the first large-scale 3D grounding dataset of RGB-D scans with object-level captions. ReferIt3D (Achlioptas et al., 2020) provides similar datasets with fine-grained object classes focusing on spatial relations of objects in a scene. Uni3DL (Xiang Li & Elhoseiny, 2023) tackles multiple 3D recognition tasks with a single model such as 3D referring segmentation (grounding), 3D captioning, classification, etc. Although these works tackle the same problem of 3D object grounding, they focus primarily on 3D tasks and design specialist models. On the other hand, **CUBE-LLM** consider image-based 3D object grounding as an extension of general image-based multi-modal large language models.

## C  DATASET DETAILS

**LV3D.** Each data in LV3D is an image and annotation pair. Each annotation consists of a list of objects present in each image. Each object has a list of question and answer pairs as described in Section 3.2 of the main paper. If the data is from 2D dataset (*e.g.*, COCO), the question answer pairs include "text → 2D box", "2D center → 2D box", "2D box → text", *etc.* Similarly, if the data is from 3D dataset (*e.g.*, NuScenes), the question includes "text → 3D box", "2D center → 3D box", "2D center → depth", "2D box → text", *etc.*, as discussed in the Section 3 of the main paper. To supplement text information, we leverage metadata from each dataset for each object class, such as object attribute in NuScenes dataset ("pedestrian" → "a walking pedestrian."). For GRIT Peng et al. (2023), we used the subset of the first 500 folders, which is about $\frac{1}{3}$. AS Wang et al. (2024a) is a collection of VQA datasets as well as some machine-generated 2D grounding data from a subset of SegmentAnything-1B Kirillov et al. (2023). The original annotations contain a substantial amount of noise with duplicate answers. We simply remove the question-answer pairs of exactly identical and irrelevant answers. We also convert all the bounding boxes to follow the same format as **CUBE-LLM**. For data standardization, we follow Omni3D Brazil et al. (2023) and convert all datasets to a virtual camera of focal length $f = 512$.

**Indoor 3D grounding benchmark.** We use the testset of Objectron Ahmadyan et al. (2021), ArkitScenes Baruch et al. (2021), and SUN-RGBD Song et al. (2015) to evaluate the 3D grounding performance of **CUBE-LLM**. In particular, we show the impact of data scaling with a smaller subset of our pre-training dataset, LV3D-small. In LV3D-small, we remove the GRIT subset Peng et al. (2023), AS-filtered Wang et al. (2024a), Waymo Sun et al. (2020), Lyft Houston et al. (2021), Argoverse2 Wilson et al. (2021), while both LV3D and LV3D-small have the same amount of indoor datasets. To evaluate grounding performance, we measure precision at $\tau$ where $\tau \in [0.15, 0.25, 0.5]$. When an image contains more than one object associated with the input text prompt, we consider the max IOU. To augment object location to the text prompt, we add "<object> close to camera" if the depth is less than 0.8m. We add "<object> on the left" or "<object> on the right" if the object center is within the left/right 20 % of the image and the distance from the

camera is 1/4/10 me away for small/medium/large objects. We define an object as small/medium/large by the max dimension $(w, h, l)$, with a threshold of 0.5, 2, 3m. Similarly, we add "`<object> at the center`" if the object center is within the center 20 % and the distance from the camera is 1/4/10 m away for small/medium/large objects.

**DriveLM-QA training.** We aim to be consistent with the baseline training recipe Sima et al. (2023). We preprocess DriveLM questions and answers to follow the bounding box format of **CUBE-LLM**; 3 decimal places, normalized between 0 and 1. For both LLaVA and **CUBE-LLM**, we train on DriveLM-QA for 5 epochs. For both LLaVA and **CUBE-LLM**, we use image resolution of $336 \times 336$ and simply fed the 6 images independently to the vision encoder and concatenated them before feeding them to the language model. The number of vision tokens is $576 \times 6$ for each frame. We do not use any additional input (*e.g.*, previous frames or point cloud) to compare to the baselines although **CUBE-LLM** can enhance 3D perception with specialists. We hold out scene IDs:

```
"64a3a2d22172406c848f2a92275808ba", "08be42eb2186411d8e2201225329f1c6",
"4b5bf3f4668d44fea9a676e9c4a8a79e", "0e247ba64b9d4a34a7256b6c173b1b5d",
"dbd9183e1278475ea54761297e004b04", "4098aaf3c7074e7d87285e2fc95369e0",
"9f3c8453d03d4df5946444757376b826", "2fc3753772e241f2ab2cd16a784cc680",
"d0880a386b6d434bb5cd13c134af7a3e", "01c3f5e39956402da3e37845632fadca"
```

in *our split* evaluation.

DriveLM dataset comprises questions about *perception* (e.g., "what are the objects worth noting in the current scenario?"), *prediction* (e.g., "Where might the van, the sedan, and the pedestrian move in the future?), *planning* (e.g., "What are the safe actions of the ego car considering those objects?") and *behavior* (e.g., "what would the ego vehicle's next action would be?").

# D   TALK2CAR GROUNDING WITH VCOT.

Figure 9 visualizes our visual chain-of-thought prompting inference on Talk2Car images. For each image and text prompt, we first ask with question:

"`Please provide 2D bounding box of the region this sentence describes: <caption>.`".

Then, with the model prediction, we construct the second question as:

"`Please provide 2D bounding box of the region this sentence describes: <caption>.`"

`<2D bounding box>`

"`Please provide 3D bounding box of the region this sentence describes: <caption>.`" This simulates multi-turn conversation and the model can attend to the tokens of the previous conversation to infer the final output. We witness that as the text prompt becomes more complicated, the guidance of the 2D bounding box helps more.

# E   DRIVELM-QA VISUALIZATION

Figure 13, 14, and 15 show various types of DriveLM questions. A large portion of the questions asks about a particular object specified in `<object ID, camera name, x, y>` format. **CUBE-LLM** is capable of reasoning about the surrounding environment from the input multi-view images. When the **CUBE-LLM** and the ground truth do not align (*e.g.*, Figure 13 top and 15 bottom), it is evident that **CUBE-LLM** understands the overall layout of surrounding objects relative to the ego vehicle. Figure 16, 17 and 7 are the QA samples specifically for grounding important objects nearby. Notable points are that some of the objects that **CUBE-LLM** predicts that do not align with the ground truth (colored in red) are still important in each driving scenario. For example, in Figure 16 **CUBE-LLM** predicts a traffic sign (warning sign for crossroad), in Figure 17 **CUBE-LLM** predicts a white sedan in front right camera that the ego may need to pay attention to, and in Figure 7 **CUBE-LLM** predicts a white sedan in back camera.

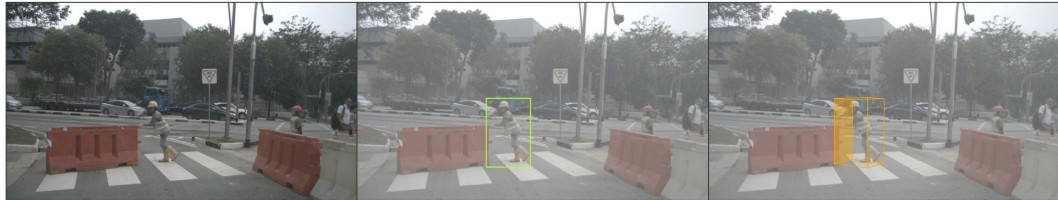

*"Stop in front of the man in the yellow hat."*

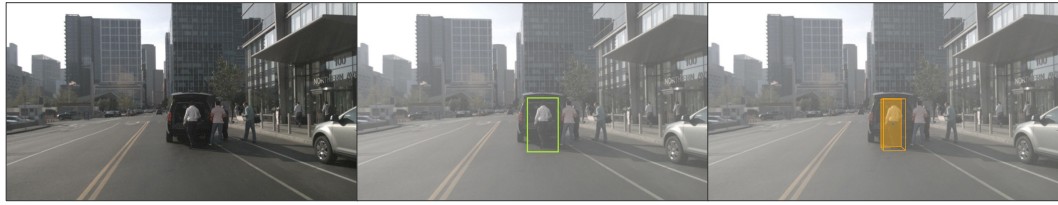

*"Pull up behind the guy wearing a white shirt. He is my uncle."*

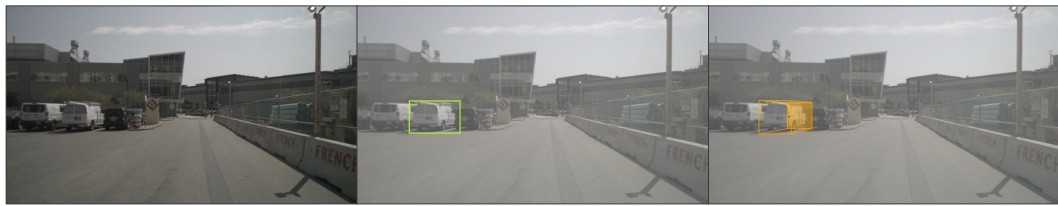

*"Turn left here, park behind the white van, left of the black car."*

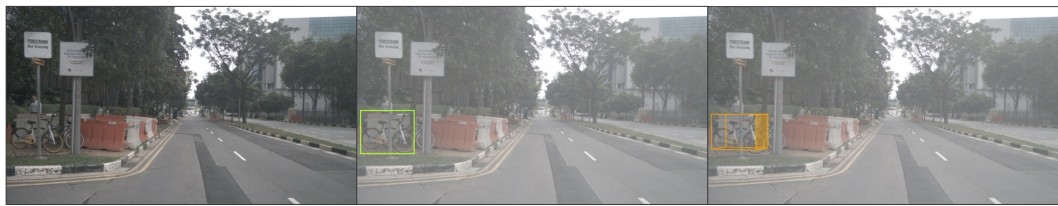

*"Stop close to this bike so I can see if it is my friend's or not."*

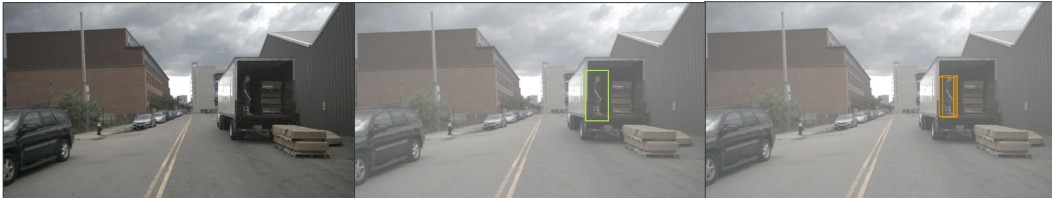

*"Stop near Joe in the truck."*

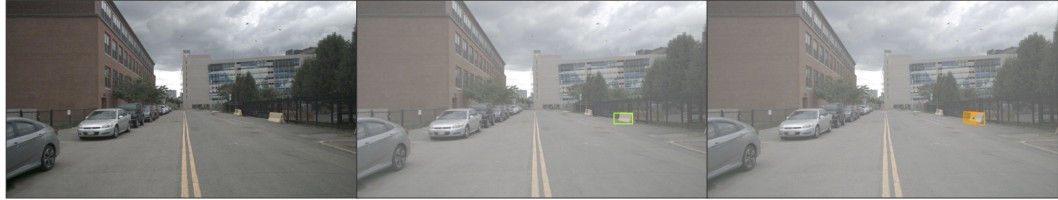

*"Pull alongside the first barrier on the right."*

Figure 9: **CUBE-LLM visual chain-of-thought prompting inference.** The first column is an input image, the second column is the 2D bounding box prediction, and the third column is the final 3D bounding box prediction prompted with the 2D prediction and text.

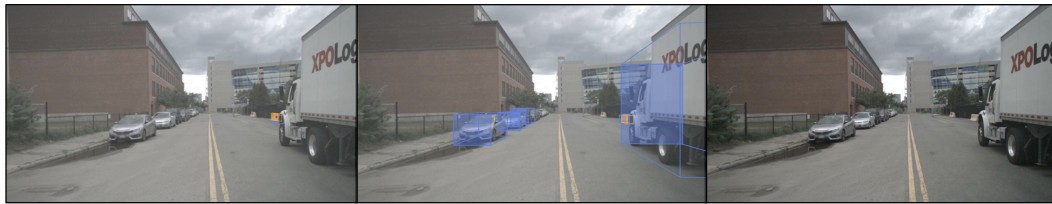

"Park near the closest concrete barrier."

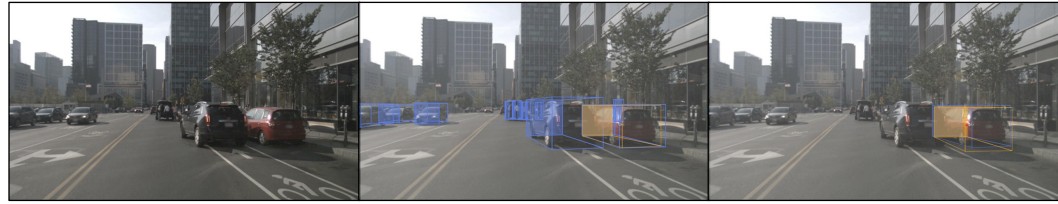

"Park behind this read car."

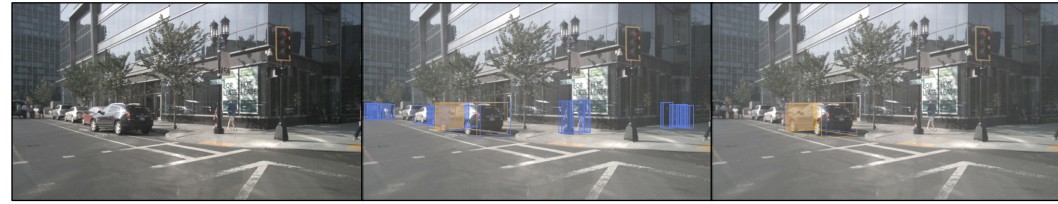

"That car is pulling out. Slow down."

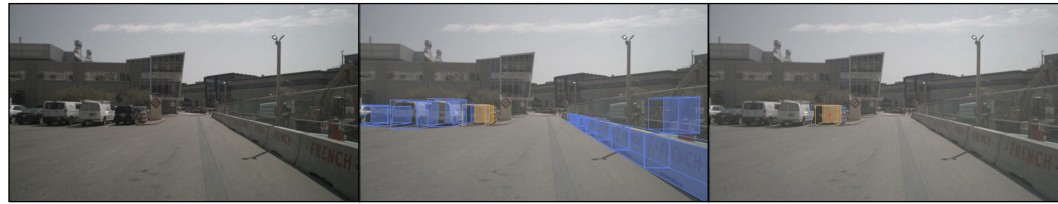

"Pull over behind that black car at the end of that row of parked  cars."

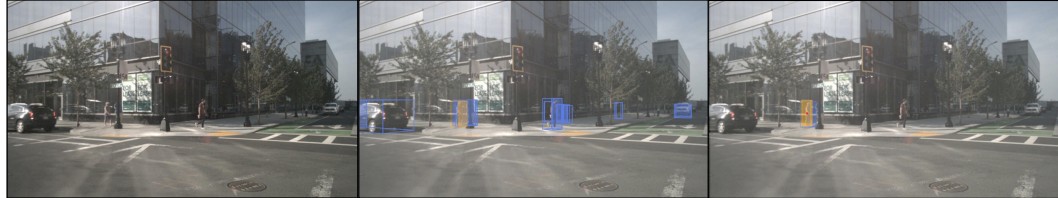

"Oh, I see my friend walking right there right next to the tree. Please stop near by her."

Figure 10: **CUBE-LLM visual prompting inference with specialist predictions.** First column is an input image, the second column is the specialist predictions (**blue**) and the ground truth (**orange**), and the third column is the final 3D bounding box prediction of **CUBE-LLM**.

## F  FAILURE CASES

In Figure 18 and 19, we show some failure cases of **CUBE-LLM** grounding result on DriveLM test set. **CUBE-LLM** makes incorrect prediction mainly in two reasons: *inaccurate depth* and *semantic mismatch*. Figure 18 shows three examples of inaccurate depth errors and Figure 19 shows three examples of semantic mismatch. Notably, for the inaccurate depth cases, the projected 3D boxes show accurate 2D localization in the image coordinate. This is because **CUBE-LLM** trains to connect its 2D understanding to 3D, as described in Section 3.3 of the main paper. For the semantic mismatch cases, **CUBE-LLM** struggles in correctly recognizing attributes when two similar objects are next to each other (*e.g.*, *silver sedan* vs. *white sedan*, *gray SUV* vs. *white SUV*). Similarly, Figure 21 and Figure 20 show the failure cases of **CUBE-LLM** on Talk2Car test set. Again, **CUBE-LLM** is still able to predict the accurate size and projected 2D box region. Figure 20 shows that

**CUBE-LLM** struggles to recognize the correct color of the car under the shade, the physical status of the black car (moving vs parked), and does not understand "*closest to the curb*."

## G    LIMITATIONS

**CUBE-LLM** has several limitations. First, **CUBE-LLM** does not employ any resampling methods Dai et al. (2023); Alayrac et al. (2022) to reduce the number of vision tokens. This will limit the model to increase the input resolution to even larger than the current $672 \times 672$ (*e.g.*, $1344 \times 1344$). **CUBE-LLM** currently only supports a single frame input. However, video input is critical to correctly recognize the dynamics of the environment. As a result, **CUBE-LLM** tends to fail to correctly predict whether an object is stationary or moving, or rely on the location of an object in the scene and infer the object's dynamics (*e.g.*, a car inside a parking space is most likely stationary). We leave these limitations for future work.

## H    SOCIETAL IMPACTS

The end results of this paper provide a foundation model for comprehensive reasoning tasks with 2D and 3D scene understanding. This is of use to a broad spectrum of applications including human-computer interaction, self-driving cars, robotics applications, and so on. In particular, it has the potential to improve the safety of these systems, as correctly grounding objects in the scene de-hallucinates the model's reasoning capability. Before deployment, appropriate safety thresholds must be cleared. Our approach does not specifically leverage dataset biases, although being a machine learning approach, it is impacted as much as other machine learning techniques.

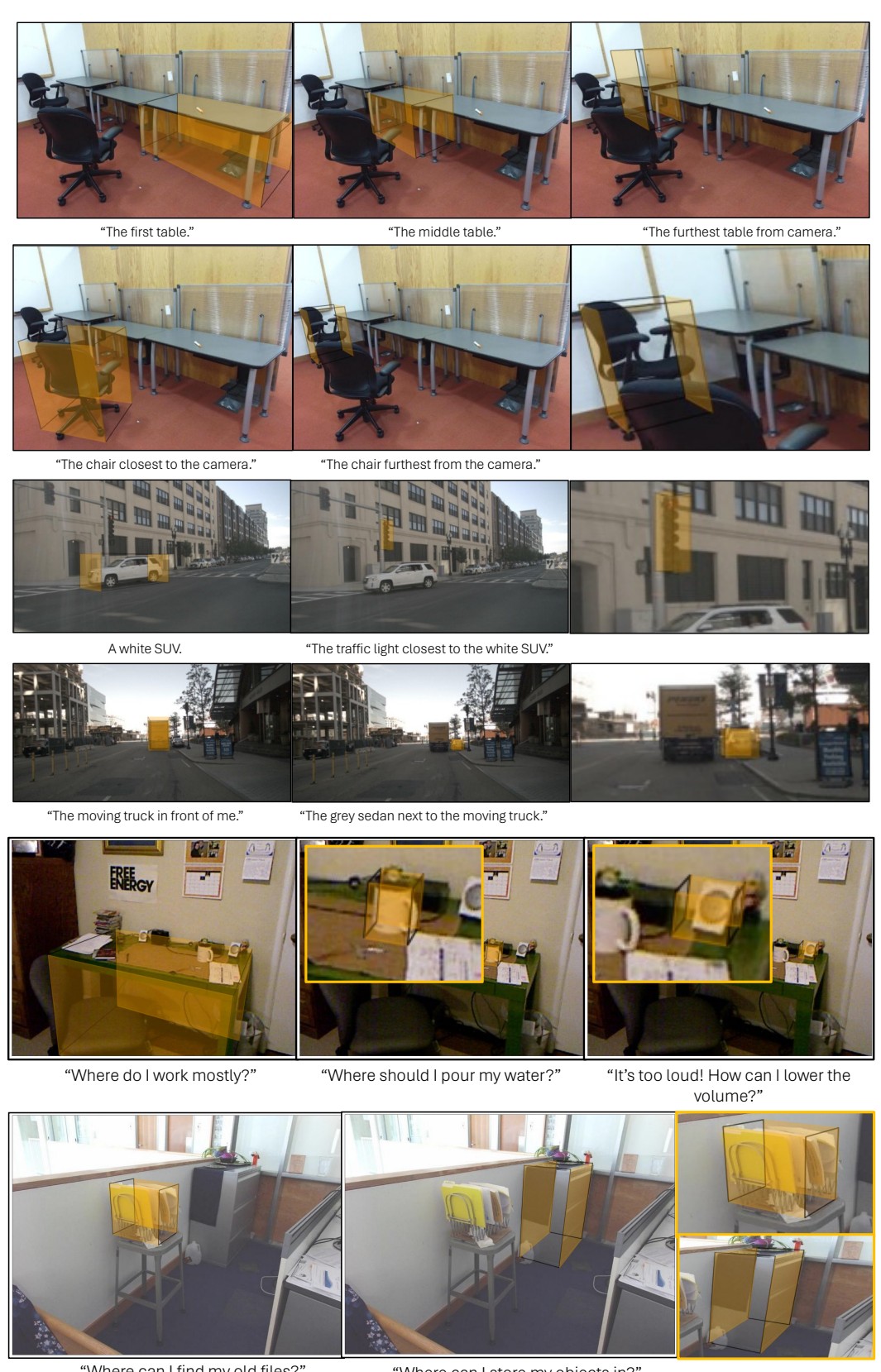

Figure 11: **More visualization of 3D grounding. CUBE-LLM** is capable of grounding object with spatial cues and understand complex questions.

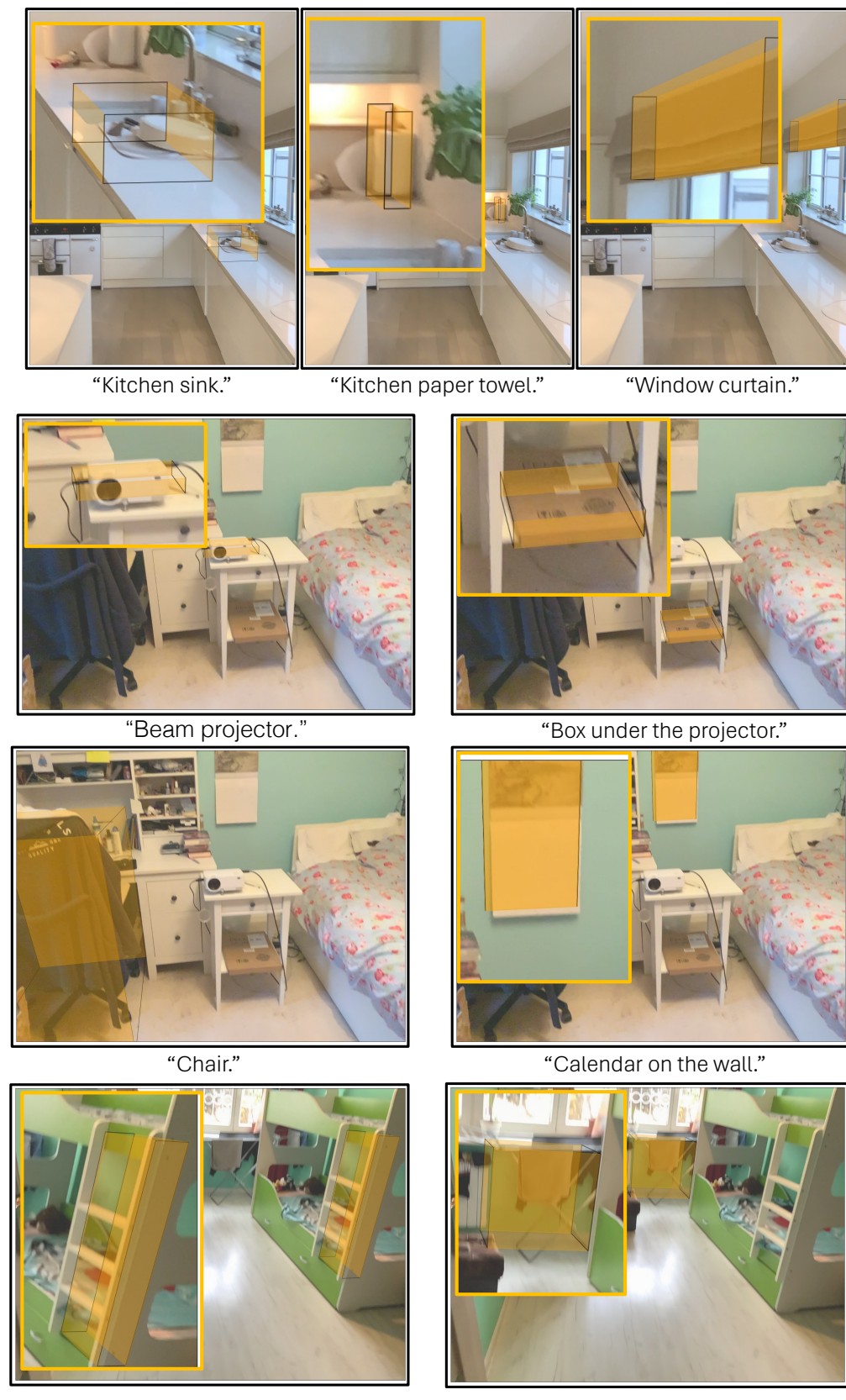

Figure 12: **More visualization of 3D grounding.** CUBE-LLM is capable of grounding open-vocabulary category names.

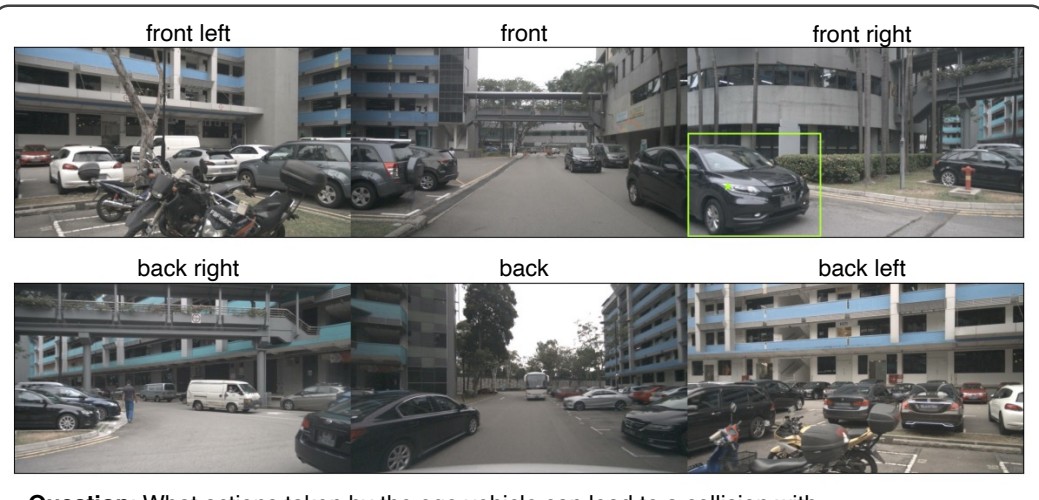

**Question**: What actions taken by the ego vehicle can lead to a collision with <c1,CAM_FRONT_RIGHT,190.8,660.8>?

**Cube-LLM**: Slight right turn.

**Ground Truth**: Moderate right turn.

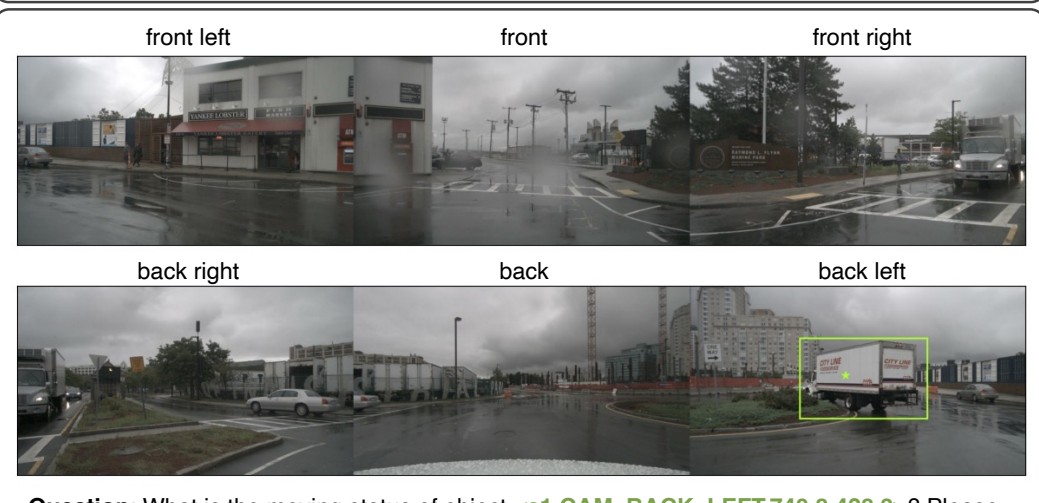

**Question**: What is the moving status of object <c1,CAM_BACK_LEFT,740.8,428.3>? Please select the correct answer from the following options: A. Back up. B. Turn left. C. Stopped. D. Turn right.

**Cube-LLM**: B. Turn left.

**Ground Truth**: B

Figure 13: **CUBE-LLM prediction on DriveLM-QA.** Green marks are the reference marks and the corresponding bounding box in the question. Orange marks are predicted 2D points by **CUBE-LLM**. Blue marks are the reference marks and the corresponding bounding box in the ground truth answers.

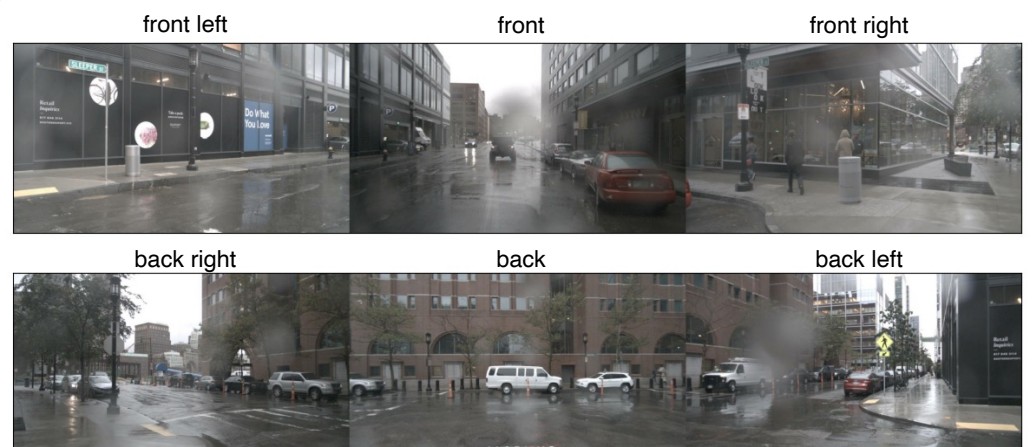

**Question**: Predict the behavior of the ego vehicle. Please select the correct answer from the following options: A. The ego vehicle is going straight. The ego vehicle is driving very fast. B. The ego vehicle is steering to the left. The ego vehicle is driving very fast. C. The ego vehicle is slightly steering to the right. The ego vehicle is driving fast. D. The ego vehicle is going straight. The ego vehicle is driving with normal speed.

**Cube-LLM**: D. The ego vehicle is going straight. The ego vehicle is driving with normal speed.

**Ground Truth**: D

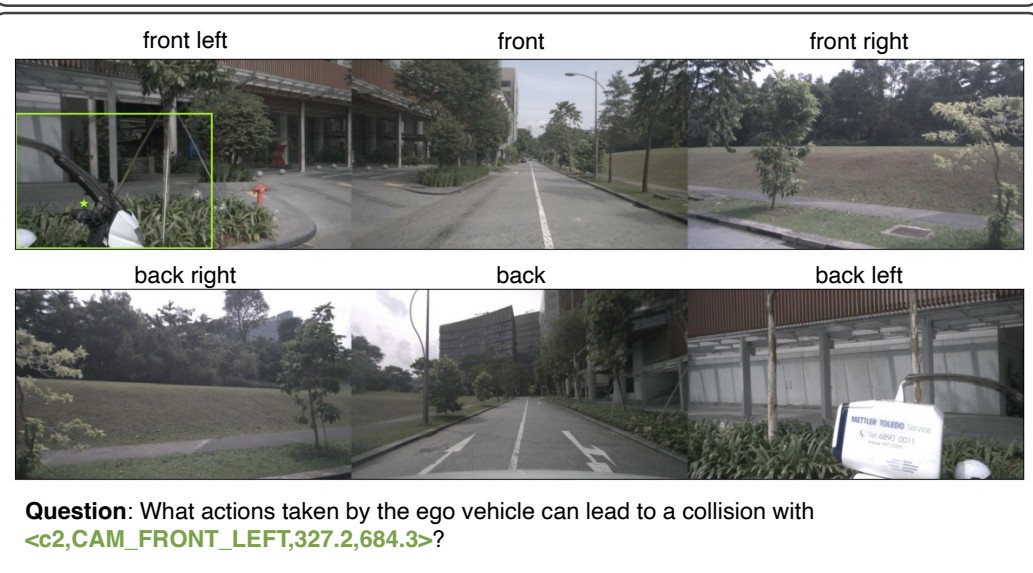

**Question**: What actions taken by the ego vehicle can lead to a collision with **<c2,CAM_FRONT_LEFT,327.2,684.3>**?

**Cube-LLM:** Slight left turn.

**Ground Truth**: Slight left turn.

Figure 14: **CUBE-LLM prediction on DriveLM-QA.** Green marks are the reference marks and the corresponding bounding box in the question. Orange marks are predicted 2D points by **CUBE-LLM**. Blue marks are the reference marks and the corresponding bounding box in the ground truth answers.

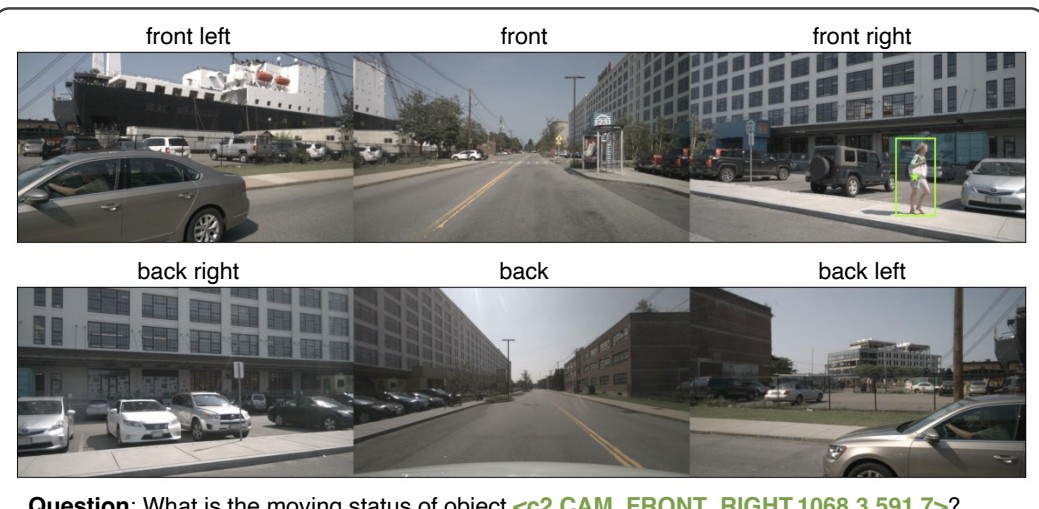

**Question**: What is the moving status of object **<c2,CAM_FRONT_RIGHT,1068.3,591.7>**? Please select the correct answer from the following options: A. Turn right. B. Stopped. C. Reverse parking. D. Going ahead.

**Cube-LLM:** D. Going ahead.

**Ground Truth**: D

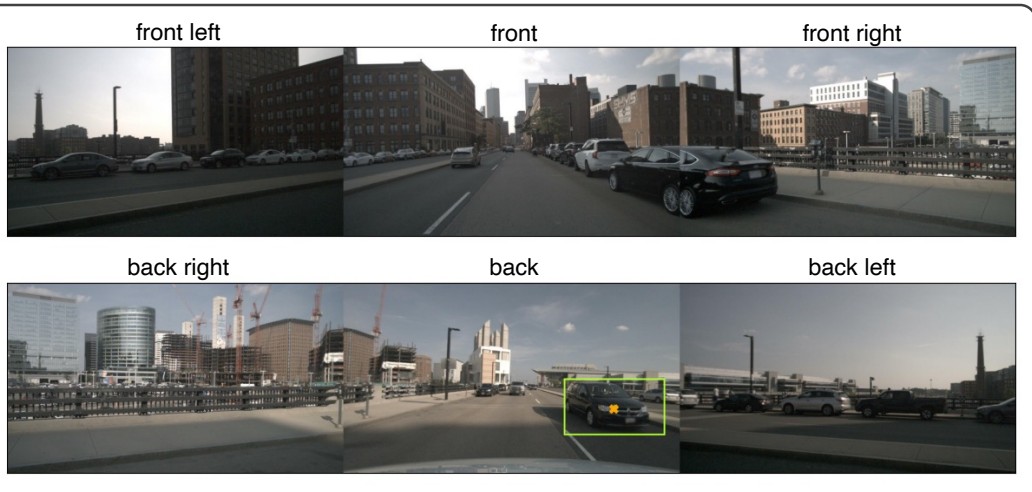

**Question**: What actions taken by the ego vehicle can lead to a collision with **<c2,CAM_BACK,1285.8,596.7>**?

**Cube-LLM:** Reversing to the left rear can lead to a collision with **<c2,CAM_BACK,1286.4,596.7>**.

**Ground Truth**: Changing to the left lane.

Figure 15: **CUBE-LLM prediction on DriveLM-QA.** Green marks are the reference marks and the corresponding bounding box in the question. Orange marks are predicted 2D points by **CUBE-LLM**. Blue marks are the reference marks and the corresponding bounding box in the ground truth answers.

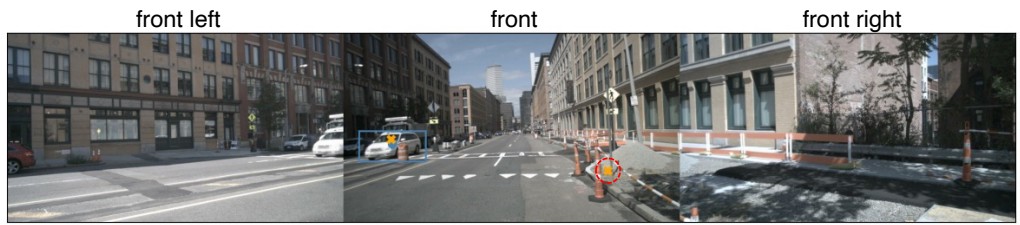
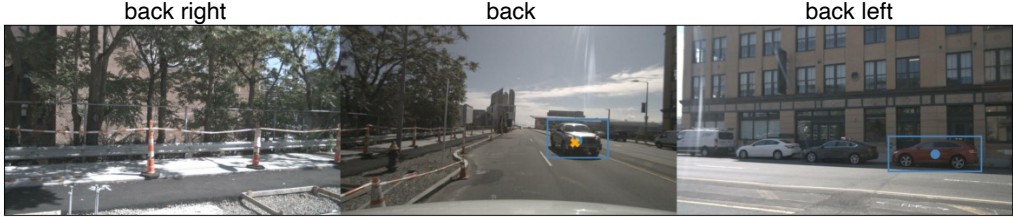

**Question**: What object should the ego vehicle notice first when the ego vehicle is getting to the next possible location? What is the state of the object that is first noticed by the ego vehicle and what action should the ego vehicle take? What object should the ego vehicle notice second when the ego vehicle is getting to the next possible location? What is the state of the object perceived by the ego vehicle as second and what action should the ego vehicle take? What object should the ego vehicle notice third? What is the state of the object perceived by the ego vehicle as third and what action should the ego vehicle take?

**Cube-LLM**: Firstly, notice that **<c3,CAM_FRONT,1257.6,656.1>**. The object is a traffic sign, so the ego vehicle should slow down and proceed. Secondly, notice that **<c1,CAM_FRONT,227.2,506.7>**. The object is going ahead, so the ego vehicle should continue going ahead at the same speed. Thirdly, notice that **<c2,CAM_BACK,1115.2,557.1>**. The object is going ahead, so the ego vehicle should continue going ahead at the same speed.

**Ground Truth**: Firstly notice that **<c1,CAM_FRONT,232.5,528.3>**. The object is going ahead, so the ego vehicle should keep going ahead at the same speed. Secondly notice that **<c2,CAM_BACK_LEFT,1228.3,611.7>**. The object is stationary, so the ego vehicle should keep going ahead at the same speed. Thirdly notice that **<c3,CAM_BACK,1126.7,549.2>**. The object is going ahead, so the ego vehicle should keep going ahead at the same speed.

Figure 16: **CUBE-LLM prediction on DriveLM-QA.** Green marks are the reference marks and the corresponding bounding box in the question. Orange marks are predicted 2D points by **CUBE-LLM**. Blue marks are the reference marks and the corresponding bounding box in the ground truth answers. Red circle is the predicted object that do not agree with the ground truth.

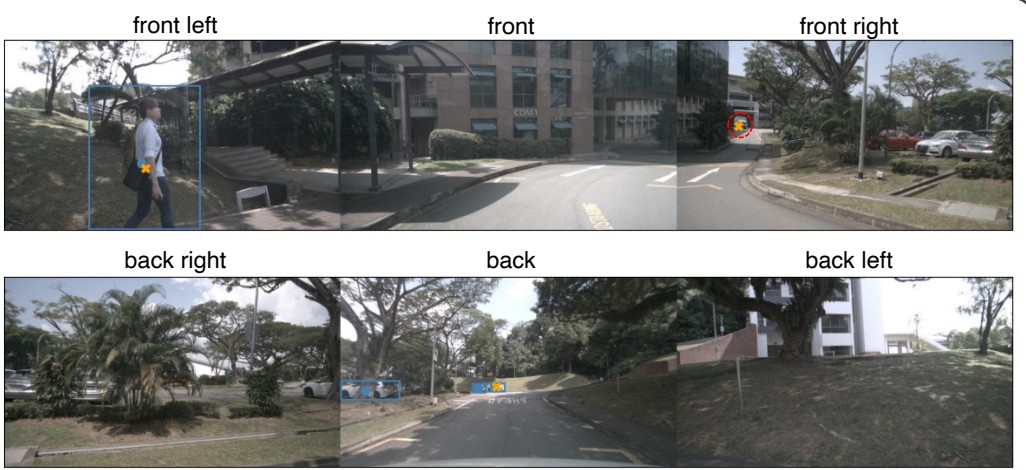

Figure 17: **CUBE-LLM prediction on DriveLM-QA.** Green marks are the reference marks and the corresponding bounding box in the question. Orange marks are predicted 2D points by **CUBE-LLM**. Blue marks are the reference marks and the corresponding bounding box in the ground truth answers. Red circle is the predicted object that does not agree with the ground truth.

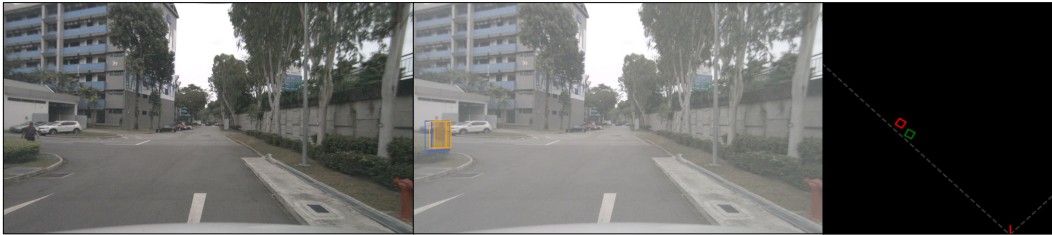

*"Elderly person in a floral shirt, moving."*

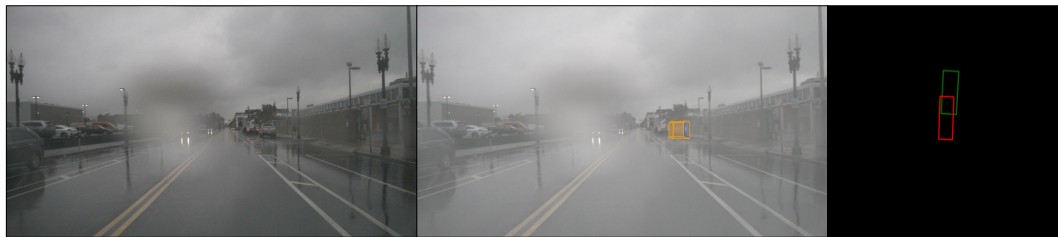

*"White pickup truck, stationary."*

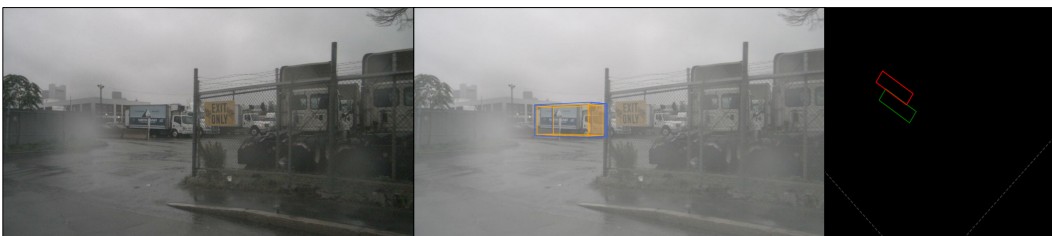

*"Blue and white truck, stationary."*

Figure 18: **Failure cases of DriveLM-Grounding images.** The error mainly attributes to incorrect depth. Each row has the original image (left), projected 3D box prediction and ground truth (middle), and BEV image (right). Blue box is the ground truth and Orange box is the prediction. In BEV images, Green box is the ground truth and red box is the prediction.

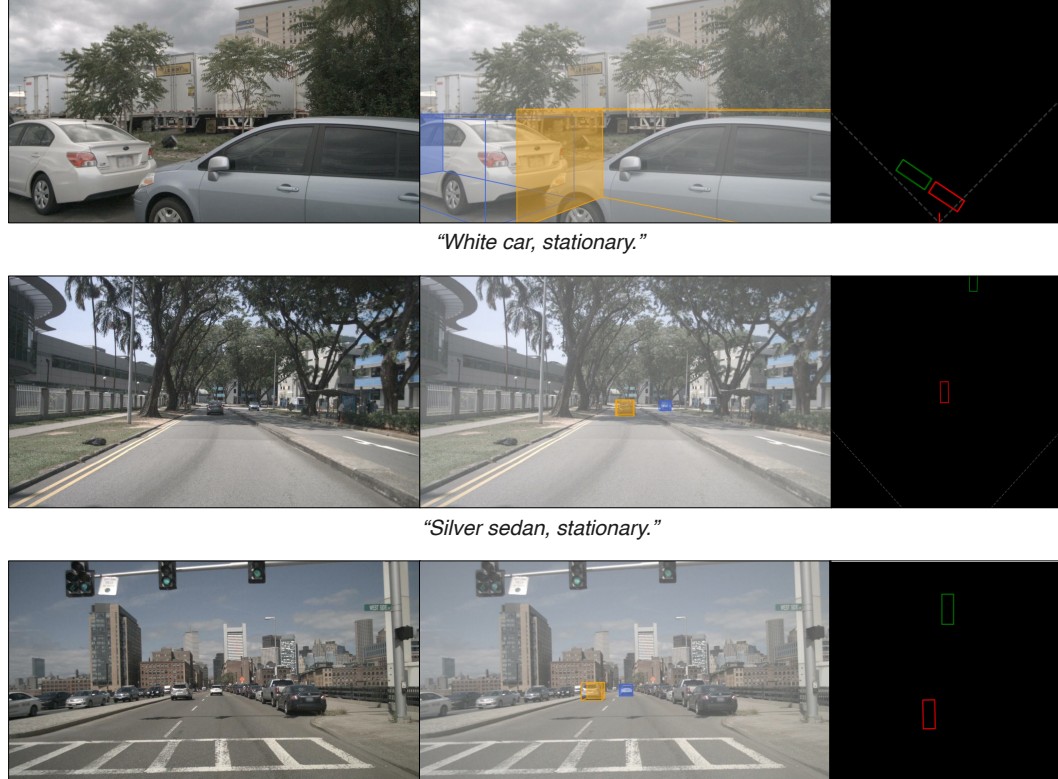

*"White car, stationary."*

*"Silver sedan, stationary."*

*"White SUV, moving."*

Figure 19: **Failure cases of DriveLM-Grounding images.** The error mainly attributes to semantic mismatch. Each row has the original image (left), projected 3D box prediction and ground truth (middle), and BEV image (right). Blue box is the ground truth and Orange box is the prediction. In BEV images, Green box is the ground truth and red box is the prediction.

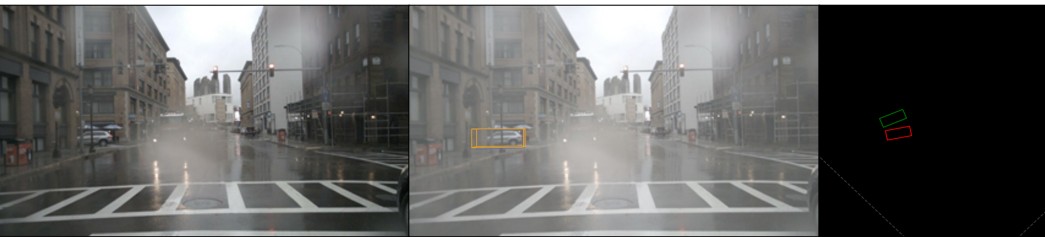

*"Once the light turns green, turn left behind the silver car."*

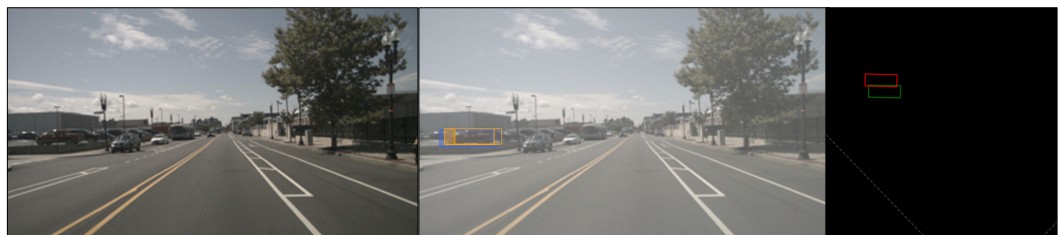

*"There is a red truck parked in a parking lot on the left hand side. Get over there."*

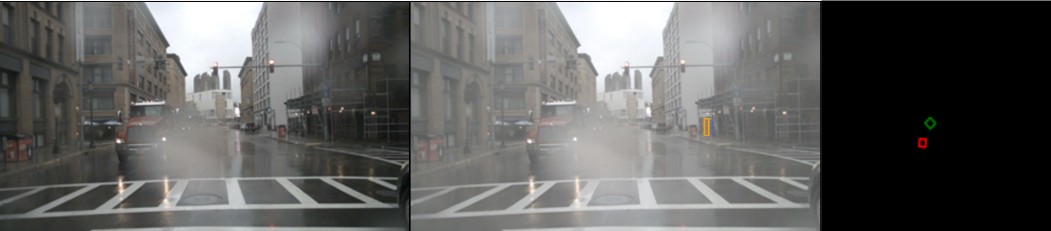

*"Stop next to my colleague who is standing on the right side of the road."*

Figure 20: **Failure cases of Talk2Car images.** The error mainly attributes to incorrect depth. Each row has the original image (left), projected 3D box prediction and ground truth (middle), and BEV image (right). Blue box is the ground truth and Orange box is the prediction. In BEV images, Green box is the ground truth and red box is the prediction.

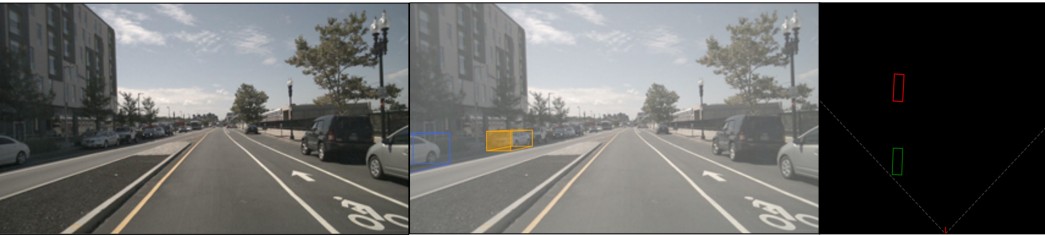

*"Once the light turns green, turn left behind the silver car."*

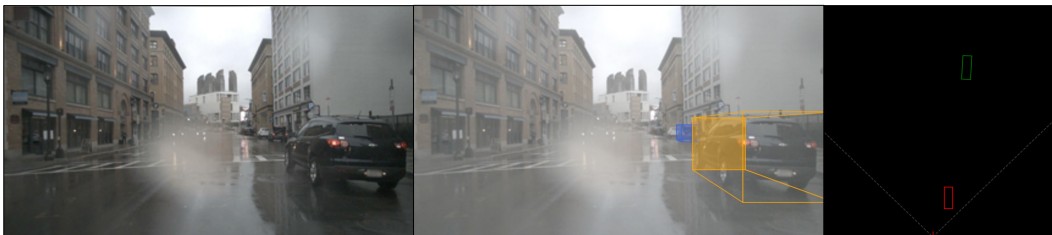

*"Switch to right lane and park on right behind parked black car."*

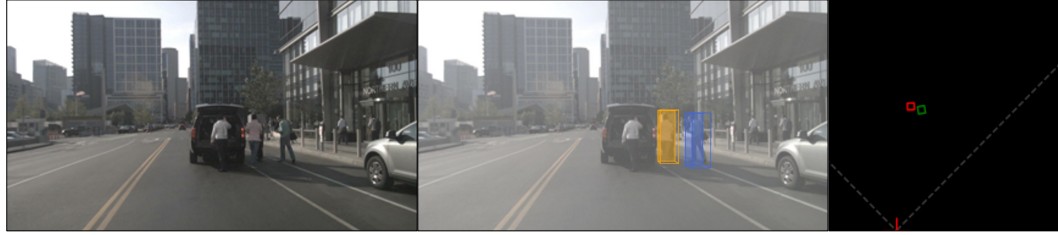

*"My friend is the guy standing closest to the curb, next to that car in front of us. Pull over so he can get in."*

Figure 21: **Failure cases of Talk2Car images.** The error mainly attributes to semantic mismatch. Each row has the original image (left), projected 3D box prediction and ground truth (middle), and BEV image (right). Blue box is the ground truth and Orange box is the prediction. In BEV images, Green box is the ground truth and red box is the prediction.

