# OpenReview forum: "Language-Image Models with 3D Understanding"
_ICLR.cc/2025/Conference — ICLR 2025 Poster_

### Official Review · Reviewer_1jbS · 2024-10-23

**Soundness:** 3
**Presentation:** 3
**Contribution:** 3
**Rating:** 6
**Confidence:** 4

**Summary:**

The paper focuses on fine-tuning LLAVA using a specially constructed 3D visual chain-of-thought (CoT) prompting data for 3D grounding tasks. The CoT data is created by arranging multiple questions about the same object in an easy-to-hard order, such as progressing from 2D box center prediction to 2D box coordinate generation, and eventually to 3D box coordinate prediction. The effectiveness of this proposed CoT data is demonstrated by training on a mixture of indoor and outdoor datasets and evaluating the results on Talk2Car and DriveLM datasets.

**Strengths:**

1. The concept of a 3D visual chain-of-thought is well-founded and logical.
2. Ablation studies effectively highlight the benefits of both the proposed training dataset LV3D and the visual chain-of-thought approach.

**Weaknesses:**

1. For the experiments, the manuscript does not provide enough comparison with recent progress in specialists, such as Grounding DINO (ECCV'24), and its follow-ups.
2. The manuscript employs a virtual camera for handling camera intrinsics. A key advantage of specialized models is their ability to process images with varying resolutions and aspect ratios once trained. Does this capability extend to virtual camera-based MLLMs? If not, what methods do you use to handle inputs with different resolutions and aspect ratios?
3. The engineering effort involved in curating data on LV3D is commendable. Do you plan to release the relevant code scripts?

Minor Typos: Inodoor -> Indoor, L432.

**Questions:**

Please see the section of weaknesses.

---

> ### Author Response · Authors · 2024-11-24
> **Response to reviewer 1jbS**
>
> We appreciate Reviewer 1jbS for the valuable feedbacks. Below we respond to each of the concerns.
>
>  > the manuscript does not provide enough comparison with recent progress in specialists, such as Grounding DINO (ECCV'24), and its follow-ups.
>
> Thank you for the reviewer's suggestion. We have added Grounding DINO L (ECCV'24) to the Table 4 of our updated manuscript for referring expression comprehension benchmark. Our Cube-LLM shows competitive result compared to Grounding DINO L (**86.6** vs **88.3**) on average score over RefCOCO/+/g metrics.
>
> > what methods do you use to handle inputs with different resolutions and aspect ratios?
>
> We preprocess all input to 672 x 672 with padding to respect different aspect ratio. This is a common practice in MLLM following Llava 1.5.
> Once the input is processed, we re-scale the camera intrinsics with respect to the image resize for calibration.
>
> > The engineering effort involved in curating data on LV3D is commendable. Do you plan to release the relevant code scripts?
>
> Thank you for your kind words regarding the engineering effort behind LV3D. We plan to release all relevant code scripts.
>
> > Minor Typos: Inodoor -> Indoor, L432.
>
> Thank you for catching the error! We have fixed in our updated manuscript (colored in red).
>
> We appreciate the positive review. If there is any further questions that would improve your assessment of our work, please feel free to let us know!

---

> > ### Comment · Reviewer_1jbS · 2024-11-24
> > **Response to the authors**
> >
> > Thank you to the authors for providing feedback that has effectively addressed my concerns. I am still inclined to recommend accepting the paper, if there are no strong objections from the other reviewers.

---

### Official Review · Reviewer_rb4W · 2024-10-29

**Soundness:** 3
**Presentation:** 3
**Contribution:** 3
**Rating:** 6
**Confidence:** 4

**Summary:**

This work presents a vision-language model designed for both 2D and 3D understanding tasks. The authors begin by reformatting existing datasets into a standardized structure, representing objects with 2D/3D bounding boxes and constructing multi-round question-answer pairs to facilitate model training. For evaluation, a chain-of-thought prompting technique is applied at the test stage, enabling the model to perform progressive reasoning on 3D tasks from simpler to more complex questions. Experiments conducted on two 3D reasoning datasets—Talk2Car and DriveLM—demonstrate promising performance for the proposed method. Additionally, the method shows competitive results on 2D vision-language (VL) tasks.

**Strengths:**

1. The exploration of LLM-based 3D reasoning is timely and relevant, given the nascent stage of 3D language-vision models compared to their 2D counterparts. One significant contribution lies in the authors’ pipeline for collecting datasets suited for training multimodal LLMs, which addresses the challenge of limited large-scale datasets for 3D tasks.

2. Experiments on two established 3D reasoning benchmarks, Talk2Car and DriveLM, show that the proposed method achieves superior performance, indicating the model’s strength in handling 3D reasoning tasks.

**Weaknesses:**

1. The collected dataset primarily focuses on 3D captioning and grounding tasks, which restricts the breadth of 3D capabilities the model can support. Expanding the dataset to include other essential 3D reasoning capabilities, such as depth ordering, neighborhood relationships, object sizing, and directionality, would better align with real-world 3D applications.

2. While the experiments indicate that the proposed method performs effectively for 3D visual grounding, it remains unclear if the model can fully comprehend and reason within the 3D environment.

3. Although this paper focuses on 3D reasoning with LLMs, it could benefit from a more thorough discussion of related recent works in 3D LLMs, such as 3D-LLM, Scene-LLM, Point-LLM, Uni3DL, and 3D visual grounding efforts, including ScanRefer and ReferIt3D.

4. In Figure 3, the input-output formats do not appear to include point_3d as a possible question or answer type, which seems inconsistent with Equations (6) and (7).

5. A recently introduced work, Cambrian, includes a 3D reasoning subset. Evaluating the proposed model on this subset could provide an insightful comparison and demonstrate the robustness of the model.

**Questions:**

It would be better to show performance on more 3D reasoning datasets/tasks.

---

> ### Author Response · Authors · 2024-11-27
> **Response to reviewer rb4W**
>
> We thank the reviewer rb4W for the valuable feedback. Below, we answer each of the reviewer’s questions.
>
> > Expanding the dataset to include other essential 3D reasoning capabilities, such as depth ordering, neighborhood relationships, object sizing, and directionality, would better align with real-world 3D applications.
>
> That is a great idea! In our task scaling in stage 2 (Figure 3), we construct different QAs that concern different 3D aspects given available labels. The reviewer's suggestion to connect this to more reasoning-related (e.g., depth ordering, object-to-object relationship, object sizing, etc.) is a great extension to our task-scaling.
>
> We had preliminary experiments with similar idea, where we construct synthetic data from object depth, relative location to camera, and relative location to objects.
> For example, we add additional spatial information such as "close by", "far away", "on left", "on right" and add it based on the object's 3D location with respect to the camera and nearby objects. (*a black sedan* $\rightarrow$ *a black sedan far away behind a pedestrian*)
> For the camera ready, we will extend this to depth ordering, object-to-object relationship and object size following the reviewer's suggestion.
>
> > More 3D reasoning benchmarks
>
> This is a valid concern. We are working to include more benchmarks that directly concerns 3D spatial reasoning such as CV-Bench from Cambrian-1 [1]. We will try our best to share the result during the rebuttal period.
>
> > More discussion of 3D LLM and 3D grounding related works
>
> We have added a section with additional related work in the updated manuscript (L850, appendix, colored in red). For the camera ready, we will refine and move them to the main paper.
>
> > Figure 3 and Eq. (6-7) are inconsistent
>
> We apologize for the confusion. The Figure 3 illustrates a *brief* summary of the task scaling. We only included a subset of tasks for illustration purpose. In practice, we randomly sampled task as the equation 6 and 7 stated.
>
> > Evaluating on Cambrian's CV-Bench
>
> Thank you for the reviewer's suggestion. We are currently working to evaluate Cube-LLM on CV-Bench [1].
>
> We thank the reviewer's valuable feedback. Please let us know if there is any remaining concerns or questions!
>
>
> [1] Cambrian-1: A Fully Open, Vision-Centric Exploration of Multimodal LLMs. *Shengbang Tong, et al. ArXiv, 2024*

---

> ### Author Response · Authors · 2024-12-02
> **Response to reviewer rb4W (2)**
>
> We completed evaluating Cube-LLM on CV-Bench [1] as mentioned in our first response. Following the reviewer rb4W's suggestion, we have compared Cube-LLM with other MLLMs for its spatial reasoning capability on CV-Bench from Cambrian-1 [1], which tests more reasoning capabilities such as *depth ordering*, *object-to-object relationship*, etc. Below we show both 2D and 3D parts of the benchmark.
>
> | **Method** | **CV-Bench$^{2D}$**| **CV-Bench$^{3D}$**|
> |:------------|:------------:|:------------:|
> |GPT-4V [2] | 64.3 | 73.8 |
> Cambrian-1-13B [1] | **72.5** | 71.8|
> |baseline | 55.5$^*$ | 49.8$^*$  |
> |SpatialRGPT [3] | 66.8$^*$  | 64.8$^*$  |
> |Cube-LLM | 67.4$^*$  | **78.8**$^*$ |
>
> Here entries with $^*$ is evaluated by us using the official code.
>
> - **Baseline** is Cube-LLM's base model before training on our LV3D dataset (i.e., stage1 model), which is comparable to LLaVA 1.5 except that the CLIP vision encoder is replaced with DINOv2.
>
> - **Cambrian-1**[1] is the MLLM from CV-Bench paper, in which perceptual capabilities are improved.
>
> - **SpatialRGPT** [3] is a concurrent work that trains an MLLM with a monocular depth estimation model and tackles spatial reasoning such as object-to-object relation.
>
> Our Cube-LLM indicates strong 3D reasoning capability compared to GPT-4V, Cambrian-1, and SpatialRGPT, improving the state of the arts by **5.0** points. Also, the improvement from baseline (**11.9** on 2D and **29.0** on 3D) highlights the impact of our LV3D dataset.
>
> Despite the promising result, there are a number of improvement we can make on Cube-LLM such as architectural improvement (e.g., efficient any resolution support or better LLM). Since Cube-LLM does not introduce any architectural specificity, we can easily plug in more advanced MLLM in our training pipeline. For the camera-ready, we will improve Cube-LLM and report these 3D reasoning benchmark as the reviewer suggested.
>
> Thank you for your great suggestion, we believe this evaluation makes our paper stronger and shows the 3D reasoning capability of Cube-LLM. Please let us know if there is any remaining questions or concerns that can improve the reviewer's assessment of our paper.
>
>
>
> [1] Cambrian-1: A Fully Open, Vision-Centric Exploration of Multimodal LLMs. *Shengbang Tong, et al. ArXiv, 2024*
>
> [2] GPT-4V System Card. *OpenAI.*
>
> [3] SpatialRGPT: Grounded Spatial Reasoning in Vision Language Models. *An-Chieh Cheng, et al. Neurips 2024.*
>
> [4] https://huggingface.co/a8cheng/SpatialRGPT-VILA1.5-8B
>
> [5] https://github.com/AnjieCheng/SpatialRGPT/blob/main/scripts/srgpt/eval/srgpt_bench.sh

---

> > ### Comment · Reviewer_rb4W · 2024-12-02
> >
> > Thanks for your response. The newly added experiment looks good.

---

### Official Review · Reviewer_Rg7K · 2024-11-02

**Soundness:** 3
**Presentation:** 3
**Contribution:** 3
**Rating:** 6
**Confidence:** 4

**Summary:**

In this work the authors proposed a method to extend vision-language models (LLaVA-v1.5) for 3D understanding. Specifically the authors constructed a large pretraining dataset LV3D (by unifying the data formats of multiple previous datasets), and a new Cube-LLM. With data and task scaling the authors show that Cube-LLM achieve improved performance for 2D and 3D grounding tasks. Results also show that Cube-LLM has some 3D reasoning capabilities.

**Strengths:**

1. The authors proposed a new MLLM for 3D understanding, *i.e.*, Cube-LLM. To enable joint learning from both 2D and 3D domain knowledge, the authors proposed datasets with unified 2D and 3D formats, as well as training tasks on different data modalities.
2. Experiments results show the benefit of the proposed data+task training paradigm, with improved performance on 2D and 3D visual grounding, as well as tasks that require certain reasoning.
3. The proposed framework will enable future research on 2D and 3D reasoning, with a unified interface of 2D and 3D representations. This can enable models to interact with 2D and 3D data, and to perform explicit reasoning on 3D layouts.

**Weaknesses:**

1. Experiments on DriveLM QA is comparing with weak baselines and the reasoning examples on DriveLM are with finetuning. For instance, how is the reasoning capabilities of the proposed Cube-LLM compared to methods with spatial reasoning training data, such as SpatialRGPT and SpatialVLM. I think this is an interesting topic to study the reasoning capabilities of the proposed method but current results are not very promising.
2. Table 4 shows that Cube-LLM outperforms previous specialist and generalist models. However, Cube-LLM is trained on RefCOCO annotations as stated in Table 1, which makes the results not a fair comparison?

**Questions:**

1. Will the datasets be released to benefit future research?

---

> ### Author Response · Authors · 2024-11-27
> **Response to reviewer Rg7K**
>
> We appreciate the reviewer Rg7K for the valuable feedback. We will respond to each of the questions and concerns below.
>
> > More 3D reasoning benchmarks
>
> That is a valid concern. We are currently working to include more benchmarks that concern 3D reasoning capability such as CV-Bench from Cambrian-1 [1]. We will try our best to share the result during the rebuttal period.
>
>
> >Cube-LLM is trained on RefCOCO annotations as stated in Table 1, which makes the results not a fair comparison?
>
> In Table 4, all specialist and generalist models train on RefCOCO datasets. The table does not include any zeroshot models.
>
> > Will the datasets be released to benefit future research?
>
> Thank you for your interest! We will release LV3D datasets and related code scripts. .
>
> We appreciate the reviewer’s valuable feedback. Please let us know if there are any additional concerns or questions that would improve the reviewer’s assessment of our work.
>
> [1] Cambrian-1: A Fully Open, Vision-Centric Exploration of Multimodal LLMs. *shengbang Tong, et al. ArXiv, 2024*

---

> ### Author Response · Authors · 2024-12-02
> **Response to reviewer Rg7K (2)**
>
> We completed evaluating Cube-LLM on CV-Bench [1] as mentioned in our first response. Following the reviewer Rg7K's suggestion, we have compared Cube-LLM with other MLLMs for its spatial reasoning capability on CV-Bench from Cambrian-1 [1], which is a reasoning task about object's 2D and 3D location relative to the camera or other objects in the scene.
>
> | **Method** | **CV-Bench$^{2D}$**| **CV-Bench$^{3D}$**|
> |:------------|:------------:|:------------:|
> |GPT-4V [2] | 64.3 | 73.8 |
> Cambrian-1-13B [1] | **72.5** | 71.8|
> |baseline | 55.5$^*$ | 49.8$^*$  |
> |SpatialRGPT [3] | 66.8$^*$  | 64.8$^*$  |
> |Cube-LLM | 67.4$^*$  | **78.8**$^*$ |
>
> Here entries with $^*$ is evaluated by us using the official code. **Baseline** is Cube-LLM's base model before training on our LV3D dataset (i.e., stage1 model), which is comparable to LLaVA 1.5 except that the CLIP vision encoder is replaced with DINOv2. For **SpatialRGPT** [3], we used ``SpatialRGPT-VILA1.5-8B'' [4] model and evaluated using the official evaluation script [5]: RGBD input from *Depth-Anything* together with mask input using the bounding boxes provided from 3D parts of CV-Bench.
>
> Our Cube-LLM indicates strong 3D reasoning capability compared to GPT-4V, Cambrian-1, and SpatialRGPT, improving the state of the arts by **5.0** points. Also, the improvement from baseline (**11.9** on 2D and **29.0** on 3D) highlights the impact of our LV3D dataset.
>
> Despite the promising result, there are a number of improvement we can make on Cube-LLM such as architectural improvement (e.g., efficient any resolution support or better LLM). For the camera-ready, we will improve Cube-LLM and report these 3D reasoning benchmark as the reviewer suggested.
>
> Thank you for your great suggestion, we believe this evaluation makes our paper stronger and shows the 3D reasoning capability of Cube-LLM. Please let us know if there is any remaining questions or concerns that can improve the reviewer's assessment of our paper.
>
>
> [1] Cambrian-1: A Fully Open, Vision-Centric Exploration of Multimodal LLMs. *Shengbang Tong, et al. ArXiv, 2024*
>
> [2] GPT-4V System Card. *OpenAI.*
>
> [3] SpatialRGPT: Grounded Spatial Reasoning in Vision Language Models. *An-Chieh Cheng, et al. Neurips 2024.*
>
> [4] https://huggingface.co/a8cheng/SpatialRGPT-VILA1.5-8B
>
> [5] https://github.com/AnjieCheng/SpatialRGPT/blob/main/scripts/srgpt/eval/srgpt_bench.sh

---

> > ### Comment · Reviewer_Rg7K · 2024-12-03
> > **Official Comment by Rg7K**
> >
> > Thanks for the responses. The new results on CV-Bench3D look good. The authors should also add the implementation and experiment details to the revision.

---

> ### Author Response · Authors · 2024-12-04
>
> Thank you for the suggestions. We will improve the implementation and experiment details for the revision.

---

### Official Review · Reviewer_vCMC · 2024-11-02

**Soundness:** 3
**Presentation:** 3
**Contribution:** 3
**Rating:** 6
**Confidence:** 4

**Summary:**

The paper introduces CUBE-LLM, extending LLAVA to 3D using the new LV3D dataset, which unifies 2D and 3D data in a question-answer format. This allows smooth 2D-to-3D generalization without changing the model’s architecture. Instead of specialized design, CUBE-LLM achieves 3D reasoning through diverse data, using iterative 2D predictions to boost 3D accuracy. It sets new benchmarks in 3D grounding on tasks like Talk2Car and DriveLM while staying competitive on standard 2D tasks.

**Strengths:**

1. The overall data pipeline is well-structured, extending LLAVA to a 3D data format with large-scale pretraining. It incorporates standardization of 2D/3D data labels (as shown in Fig. 3), unification of model I/O inputs, and the implementation of visual chain-of-thought (CoT) reasoning for step-by-step analysis.

2. The evaluation is comprehensive, covering various tasks such as 3D grounding (Talk2Car, DriveLM-grounding, Indoor Objectron, ARKitScenes, SUN-RGBD), QA (DriveLM-QA), 3D grounding and captioning (LV3D), and 2D grounding (RefCOCO).

3. A good portion of the visualizations effectively enhances understanding across all tasks.

4. Visual CoT achieves a 3.2-point improvement.

**Weaknesses:**

1. The paper lacks an in-depth analysis of joint 2D and 3D training. It closely follows LLAVA1.5, with DINOv2 as the vision model and primarily contributes by consolidating 2D and 3D datasets. While 3D performance improvements could add value, the impact seems insufficient for acceptance. I would like more analysis on the effects of excluding 2D and 3D box pretraining on 3D/2D QA/grounding performance.

2. In my view, the Visual CoT would benefit from zooming in on selected parts (SoM [1]) to enhance model understanding with enlarged object details in a second stage. While the 3.2-point performance improvement (Line 423-424) is reasonable, it comes at the cost of additional tokens and computation, similar to test-time augmentation.

[1] Yang, Jianwei, et al. "Set-of-mark prompting unleashes extraordinary visual grounding in gpt-4v." arXiv preprint arXiv:2310.11441 (2023).

**Questions:**

See the weakness.

A few corrections:
Line 422: Change "Table 5" to "Figure 5."
Figure 5 (bottom right): Why is the performance bar at 32% different between the two sub-figures?

**Details Of Ethics Concerns:**

N/A.

---

> ### Author Response · Authors · 2024-11-24
> **Response to reviewer vCMC**
>
> We thank the reviewer vCMC for the valuable feedback. Below, we respond to each of the questions and concerns.
>
> > more analysis on the effects of excluding 2D and 3D box pretraining on 3D/2D QA/grounding performance.
>
> The Table 2 (b) contains the exact ablations on 3D Grounding task on DriveLM dataset. Every line of Table 2 (b) is an individually pre-trained model under different conditions (the ones above included). We start from LLaVA 1.5 with a CLIP vision encoder, replacing the encoder from CLIP to DINOv2, including **2D portions of LV3D**, and including **all (2D+3D) LV3D**. Table 3 shows the 3D grounding results of indoor scenes with different amounts of 3D data, noted as LV3D-small (LV3D without a majority of outdoor 3D data). In addition, Figure 5 (top right) shows the scaling behavior of the LV3D dataset on (zeroshot) 3D grounding performance with increasing amount of pretraining data. All these results analyze and ablate the impacts of 2D and 3D components of LV3D.
>
> > in-depth analysis of joint 2D and 3D training.
>
> Although the first two stages of pre-training follow LLaVA 1.5, we design effective 2D and 3D training with stage-wise pre-training. In our updated manuscript (Figure 8, page 15 of supplementary) we illustrate the complete Cube-LLM training pipeline. The impact of this staged pre-training (*low-resolution 2D* $\rightarrow$ *high-resolution 2D+3D*) is shown in Figure 5 (bottom right, second plot) of the main paper, resulting in **8.7** points improvement in zeroshot 3D grounding task.
>
> Moreover, we believe that the contribution of Cube-LLM is the simplification of training 3D understanding *without any degradation in overall MLLMs*. Training 3D typically requires 3D-specific designs and our goal is to train it on any MLLM as an *addition* of capability.
>
> > the Visual CoT would benefit from zooming in on selected parts (SoM [1]) to enhance model understanding with enlarged object details in a second stage.
>
> This is a great idea! We agree with the reviewer that the Visual CoT will benefit from zooming in on selected parts. We did consider this exact extension of our Visual CoT during the development of Cube-LLM. However, for an MLLM to reason with *multiple images* at once (original and zoomed-in), the model requires training with interleaved image-text pair datasets as in the state-of-the-art MLLMs train with interleaved data samples (e.g., VILA[1] or MM-1[2]). Our Cube-LLM currently only supports a single image input without finetuning. For the camera-ready, we will improve our MLLM to support interleaving image-text input and experiment with the reviewer’s suggestion.
>
> In addition, we also included a brief discussion about SoM in the related work section of the updated manuscript (colored in red).
>
> > it comes at the cost of additional tokens and computation, similar to test-time augmentation.
>
> We agree to the reviewer that the improvement comes with additional computation cost. We would like to argue that improving the performance of MLLMs with test-time computation is a highly promising direction (OpenAI o1 [3]). We believe that our *Visual CoT* and *Specialist Prompting* show possibility of incorporating additional tokens (its own prediction or tokenized specialist predictions) on general MLLMs at test-time.
>
> We appreciate the reviewer's valuable feedback. Please let us know any further questions or concerns that would improve the reviewer's assessment of our work.
>
>
>
> [1] VILA: On Pre-training for Visual Language Models. *Ji Lin, et al. ArXiv, 2023.*
>
> [2] MM1: Methods, Analysis & Insights from Multimodal LLM Pre-training. *Brandon McKinzie, et al. ECCV 2024.*
>
> [3] OpenAI o1. *O1 team, OpenAI. Blog.*

---

> > ### Comment · Reviewer_vCMC · 2024-11-25
> > **Thanks for the author feedback.**
> >
> > Thank you for the authors' feedback, which addresses my concerns regarding the effects of 2D/3D pre-training. Given the straightforward and effective adaptation of llava to 3D contexts, I am increasing my rating to 6.

---

### Meta-Review · Area_Chair_MenG · 2024-12-13

**Metareview:**

This paper augments LLAVA with 3D understanding using a new LV3D dataset, which unifies the data formats of multiple previous datasets. By scaling data and tasks the authors show that their method achieves an improved performance for 2D and 3D grounding tasks. The strengths of the paper are a new dataset 2D/3D grounding and a method, CUBE-LLM for joint learning from both 2D and 3D annotations and the experiments that demonstrate the benefits of the proposed methods on various datasets. The results on open-vocabulary understanding, reasoning and 3D spatial understanding are convincing and demonstrate the benefit of integrating 3D understanding into VLMs. Hence the paper should clearly be accepted.

**Additional Comments On Reviewer Discussion:**

All reviewer concerns were addressed in the rebuttal.

---

### Decision · Program_Chairs · 2025-01-22

Accept (Poster)